# In vivo screening identifies *GATAD2B* as a metastasis driver in *KRAS*-driven lung cancer

Caitlin L. Grzeskowiak[1], Samrat T. Kundu[2], Xiulei Mo[3], Andrei A. Ivanov[3], Oksana Zagorodna[1], Hengyu Lu [1], Richard H. Chapple[1], Yiu Huen Tsang [1], Daniela Moreno[1], Maribel Mosqueda[4], Karina Eterovic[4], Jared J. Fradette[2], Sumreen Ahmad[2], Fengju Chen[5], Zechen Chong[6], Ken Chen [6], Chad J. Creighton[5,6,7], Haian Fu [3], Gordon B. Mills[4], Don L. Gibbons[2,8] & Kenneth L. Scott[1]

Genetic aberrations driving pro-oncogenic and pro-metastatic activity remain an elusive target in the quest of precision oncology. To identify such drivers, we use an animal model of *KRAS*-mutant lung adenocarcinoma to perform an in vivo functional screen of 217 genetic aberrations selected from lung cancer genomics datasets. We identify 28 genes whose expression promoted tumor metastasis to the lung in mice. We employ two tools for examining the *KRAS*-dependence of genes identified from our screen: 1) a human lung cell model containing a regulatable mutant *KRAS* allele and 2) a lentiviral system permitting co-expression of DNA-barcoded cDNAs with Cre recombinase to activate a mutant *KRAS* allele in the lungs of mice. Mechanistic evaluation of one gene, *GATAD2B*, illuminates its role as a dual activity gene, promoting both pro-tumorigenic and pro-metastatic activities in *KRAS*-mutant lung cancer through interaction with *c-MYC* and hyperactivation of the *c-MYC* pathway.

[1] Department of Molecular and Human Genetics, Baylor College of Medicine, Houston, TX 77030, USA. [2] Department of Thoracic/Head and Neck Medical Oncology, The University of Texas MD Anderson Cancer Center, Houston, TX 77030, USA. [3] Department of Pharmacology and Emory Chemical Biology Discovery Center, Emory University School of Medicine, Atlanta, GA 30322, USA. [4] Department of Systems Biology, University of Texas MD Anderson Cancer Center, Houston, TX 77030, USA. [5] Dan L. Duncan Cancer Center, Baylor College of Medicine, Houston, TX 77030, USA. [6] Department of Bioinformatics and Computational Biology, The University of Texas MD Anderson Cancer Center, Houston, TX 77030, USA. [7] Department of Medicine, Baylor College of Medicine, Houston, TX 77030, USA. [8] Department of Molecular and Cellular Oncology, University of Texas MD Anderson Cancer Center, Houston 77030, USA. These authors contributed equally: Caitlin Grzeskowiak, Samrat T. Kundu. These authors jointly supervised this work: Don L. Gibbons, Kenneth L. Scott. Deceased: Kenneth L. Scott. Correspondence and requests for materials should be addressed to D.L.G. (email: dlgibbon@mdanderson.org)

Non-small cell lung cancer (NSCLC) is the leading cause of cancer mortality in the United States, primarily due to the development of metastatic disease[1]. Recent progress in treating lung cancer has come from identifying patient sub-populations with identifiable oncogenic mutations that can be targeted with small molecule inhibitors (e.g., erlotinib and crizotinib for *EGFR*-driven and *EML4-ALK*-driven tumors, respectively)[2,3]. Unfortunately, the majority of lung cancer cases are driven by unknown genetic events or mutations in genes such as *KRAS* (30% of patients) for which there are no selective therapeutics[4].

*KRAS*-driven lung cancers are not targetable by currently approved therapies and represent a particularly aggressive form of NSCLC[5]. Regional or distant metastases are thought to form often and early in *KRAS*-driven adenocarcinoma leading to high mortality[6,7]. Subgroups have been identified in *KRAS*-mutant NSCLC based upon co-mutations that enhance or modulate *KRAS* tumorigenicity and disease progression, providing biological and molecular context for personalized and targetable treatments for *KRAS* mutant patients[8]. New approaches aimed at personalized therapeutic strategies are critical for these patients, as identifying targets downstream of or that work in conjunction with *KRAS* offer the most promising opportunities to exploit therapeutic vulnerabilities. Therefore, systematic functional characterization of lung cancer genome datasets is needed. The Cancer Genome Atlas (TCGA) and others have generated a compendium of genomic aberrations in lung cancer with the goal of identifying the most promising drug targets and diagnostic biomarkers[9]. The challenge now is to distinguish the subsets of functional oncogenic and metastatic driver aberrations from passenger mutations that do not offer therapeutic opportunities.

While RNA interference (RNAi)-based and CRISPR/Cas9-based genetic screening platforms have successfully identified new tumor suppressor genes and other genetic vulnerabilities in cancer, several recent studies reveal a complementary approach through developing scalable gain-of-function screening systems for validating over-expressed or mutationally activated oncogenes that, as a class, have served as successful therapeutic targets to date. For example, we previously reported a multi-level functional assessment platform, High-Throughput Mutagenesis and Molecular Barcoding (HiTMMoB), which has identified novel variants of known oncogenes[10], as well as elucidating novel drivers of pancreatic ductal adenocarcinoma[11]. Here we report an adaptation of this platform to identify genetic drivers that synergize with mutant *KRAS* to advance tumor progression and metastasis in lung adenocarcinoma. In vivo functional screening of a gene library informed by oncogenomics-guided integration of mutant *KRAS*-specific mouse and human gene signatures reveal several genes whose expression promote tumor growth and/or metastasis, as outlined here and in the companion paper. Among those genes, functional characterization of *GATAD2B* illuminates its role as a potent driver of tumor growth and metastasis in *KRAS*-driven lung cancers. We further show here that high *GATAD2B* expression correlates with worsened outcomes in lung cancer patients and cooperates with *KRAS* to promote gain-of-function pro-oncogenic and pro-metastatic transcriptional programs including *MYC* to mediate cell invasion in vitro and tumor progression in vivo.

## Results

### In vivo screening for drivers of lung cancer metastasis.
We and others have shown that mouse and human tumors of diverse tissue lineages sustain orthologous genomic aberrations that can function as bona fide cancer driver events[12–16]. These observations prompted us to utilize cross-species, integrative analyses to biologically filter and prioritize TCGA data to define a gene list enriched for lung cancer drivers. To do this, we first leveraged published transcriptome comparisons of spontaneous lung adenocarcinomas and metastases isolated from $Kras^{LA1/+}$, $p53^{R172H\Delta G/+}$ mice[17] in order to select 615 genes ($p < 0.01$, paired $t$-test) up-regulated in $Kras^{LA1/+}$, $p53^{R172H\Delta G/+}$ metastases compared to primary tumors. We next intersected these data with transcriptome comparisons of non- and strongly metastatic murine 393 P and 344SQ syngeneic tumors, respectively, grown from cell lines isolated from spontaneous $Kras^{G12D}$, $p53^{R172H\Delta G}$ tumors[18], which revealed 1220 genes expressed significantly higher in metastatic 344SQ cells vs. 393 P tumors ($p < 0.01$, paired $t$-test)[18]. To control for the possibility of cooperation between $KRAS^{G12D}$ and $p53^{R172H\Delta G}$ used in our model, we compared gene expression levels of all ORFs used in our screening strategy between murine lung tumors from the $KRAS^{G12D}$ and $KRAS^{G12D}$; $p53^{R172H\Delta G}$ models, and observed that less than 3% of the 225 ORFs were significantly expressed ($p < 0.01$, paired $t$-test) in the $KRAS^{G12D}$;$p53^{R172H\Delta G}$ model compared to the $KRAS^{G12D}$ model (Supplementary Data 1). Genes found up-regulated in *KRAS* models were next triangulated with human copy number amplifications documented by TCGA (1.5-fold somatic amplification across ≥5% of 154 analyzed lung adenocarcinoma specimens). In total, this cross-species analysis generated a list of 220 overlapping genes.

Because oncogenic pathways can be similarly activated by hyperactivation mutations, an observation first discovered by well-characterized oncogenes such as *PIK3CA* and *ALK*[19–23], we integrated sequencing results by Ding and colleagues[24] that focused on 623 genes with potential relationships to cancer. Filtering copy number amplifications from TCGA with the 1013 non-synonymous somatic mutations identified by their study revealed 31 amplified genes that contain at least one validated missense mutation within the 188 analyzed tumors. Combining these data with the 220 genes described above yielded a total of 251 candidate genes that include several known to play a significant role in lung cancer (e.g., *IKK-BETA*, *KRAS*, *EGFR*, *ZEB1*, *TWIST1*), in addition to two potent drivers (*FSCN1* and *HOXA1*) of melanoma transformation and metastasis identified in our previous studies[12,25]. Of the 251 candidate genes identified, 225 were available in our collection of open reading frame (ORF) clones comprised of the Human ORFeome collection[26], as well as other commercially available ORF clones. Conservation analysis was performed to reduce the potential for false negatives in our screening model and confirmed significant homology of the 225 genes between mouse and human (Supplementary Data 1). Included in the 225 ORFs were 28 wild-type ORFs representing mutated genes in Ding et. al studies (Supplementary Data 1). Each ORF was introduced into a lentiviral vector through recombination-mediated cloning. Importantly, each expression vector was uniquely tagged with a 24-nucleotide DNA barcode (Supplementary Data 1) during ORF lentiviral integration using our previously reported High-Throughput Mutagenesis and Molecular Barcoding (HiTMMoB) strategy[11], providing a surrogate identifier for each associated ORF.

We next sought to examine tumor growth and metastatic activity of selected candidates by performing a pooled, in vivo screen for drivers of lung metastasis (Fig. 1). Barcoded lentiviral constructs were used to generate lentivirus in a multi-well format, followed by transduction of non-metastatic murine lung cancer cells (393 P) derived from the $Kras^{LA1/+}$;$p53^{R172H\Delta G}$ genetically engineered mouse (GEM) model[18] from which the candidate gene library was partially derived. Of the 225 transduced cell lines, 217 positively selected for viral integration with puromycin (Supplementary Data 1). The 217 cell lines were individually transduced to reduce biases of multi-gene interactions. While

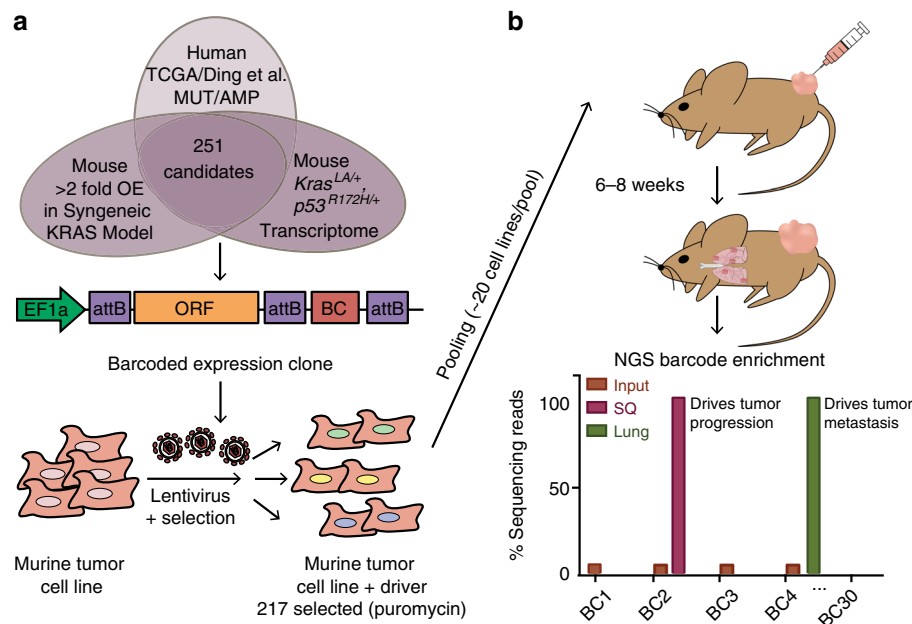

**Fig. 1** In vivo functional screening for *KRAS* effectors driving metastasis. **a** Illustration of Oncogenomics informed screening of DNA barcoded candidate genes cloned by HiTMMoB and introduced into non-metastatic 393 P murine tumor cell line via lentiviral transduction. **b** Individual ORF-barcoded cell lines were pooled and injected into the flanks of immunocompetent 129 Sv mice, followed by barcode enrichment analysis of injected cells (Input), subcutaneous (SQ) tumors and lung metastases (see text for details). Representative histogram illustrates positive enrichment of hypothetical driver of SQ growth (BC2) and lung metastasis (BC4). attB recombination cassettes; BC barcode. Elements of image used with permission of Patrick J. Lynch and Carl Jaffe, MD under Creative Commons Attribution 2.5 License 2006

individually transduced cell lines may limit complexity of the screening library, it does ensure expression of all potential oncogenes with a readily identifiable ORF-surrogate using the DNA barcode. The 217 cell lines, along with mCherry-expressing control cells, were pooled (average pool size = 20 genes/pool; Supplementary Data 2) for subcutaneous (SQ) injection into the flanks of immunocompetent syngeneic 129 Sv mice ($N = 10$ mice/pool; $N = 120$ mice total). Murine 393 P cells develop primary SQ tumors over the course of 6–8 weeks when injected into 129 Sv mice and do not metastasize as parental cells or those expressing mCherry control (Supplementary Fig. 1). In contrast, lung metastases were readily observed when animals were injected with cells across multiple pools, generating a streamlined approach for rapid analysis of metastatic lesions in our screen (Supplementary Fig. 1, Supplementary Table 1).

We have observed from in vivo tumor screens that, relative to injected cells (input), tumors and metastases positively select ORF drivers and lose those with no role in progression[10,11]. Individual enrichment among pooled ORFs can be determined by comparing the number of individual ORF-associated barcode reads within experimental pools as a ratio of each barcode to the total number of barcode reads per NGS amplicon. For example, ORF-driven SQ tumors (barcode 2) and lung metastases (barcode 4) will be positively enriched for driver-associated barcodes and lose those with no role in progression (that is, passengers, barcode 1 and 3; Fig. 1). To determine which ORFs were positively selected in tumor and metastatic tissues, we next collected quadruplicate SQ tumor and independent metastatic lung lesions when present from SQ tumor-bearing mice (Supplementary Table 1). We quantitated ORF barcode reads from genomic DNA isolated from SQ tumors, metastases, and injected cells using next generation sequencing (NGS) as described previously[11]. Our assessment of ORF barcode enrichment in SQ tumors vs. input identified 21 (representing 19 unique genes) of 217 ORFs (% of mice where $N > 2$ mice analyzed, SQ BC enrichment > input = 100×;

Supplementary Fig. 2a and Supplementary Data 2), suggesting that these ORFs may enhance cell proliferation and/or tumor growth in 393 P cells. Assessment of barcode enrichment in lung metastases vs. input identified 28 ORFs positively enriched in multiple ($N \geq 2$) lung metastasis samples in two or more mice per cohort of 10 pool-injected mice, therefore 89% of cell lines (189 in total) were not present in macrometastases collected upon necropsy (Fig. 2a and Supplementary Table 1). No macrometastases were present in cohorts injected with pools 1 and 2. Notable ORFs enriched in metastatic tissues include *GNAS* and *MYC*, which were previously shown to promote *KRAS*-driven lung cancer metastasis[27,28]. Comparison of SQ-enriched and metastasis-enriched ORFs found 7 positively enriched in both, suggesting promotion of tumor growth and metastasis, whereas 21 ORFs were found enriched only in metastases (Supplementary Fig. 2b).

**GATAD2B expression correlates with poor survival.** We sought to identify progression mediators of *KRAS*-driven lung cancer, thus we resourced TCGA datasets to determine whether amplification and copy number-driven expression (Supplementary Fig. 3a) of the 28 metastasis-enriched genes from the screen correlated with *KRAS* mutation status in lung adenocarcinoma patients. We identified three genes (*GATAD2B*, *ACVR1B*, and *ZC3H11A*) whose overexpression significantly correlates with mutant *KRAS* lung adenocarcinoma vs wild-type *KRAS* status in patients (Supplementary Table 2). Additional mining of TCGA genomics data revealed that *GATAD2B* is also commonly amplified in *KRAS* mutant cancers in other lineages, including pancreatic ductal adenocarcinoma, where *KRAS* is mutationally activated in greater than 90% of cases (Fig. 2b)[4,9,29,30]. We queried TCGA lung adenocarcinoma patient datasets fully annotated for gene expression (TCGA-provisional as queried using [http://www.cbioportal.org/])[31]. Through evaluating

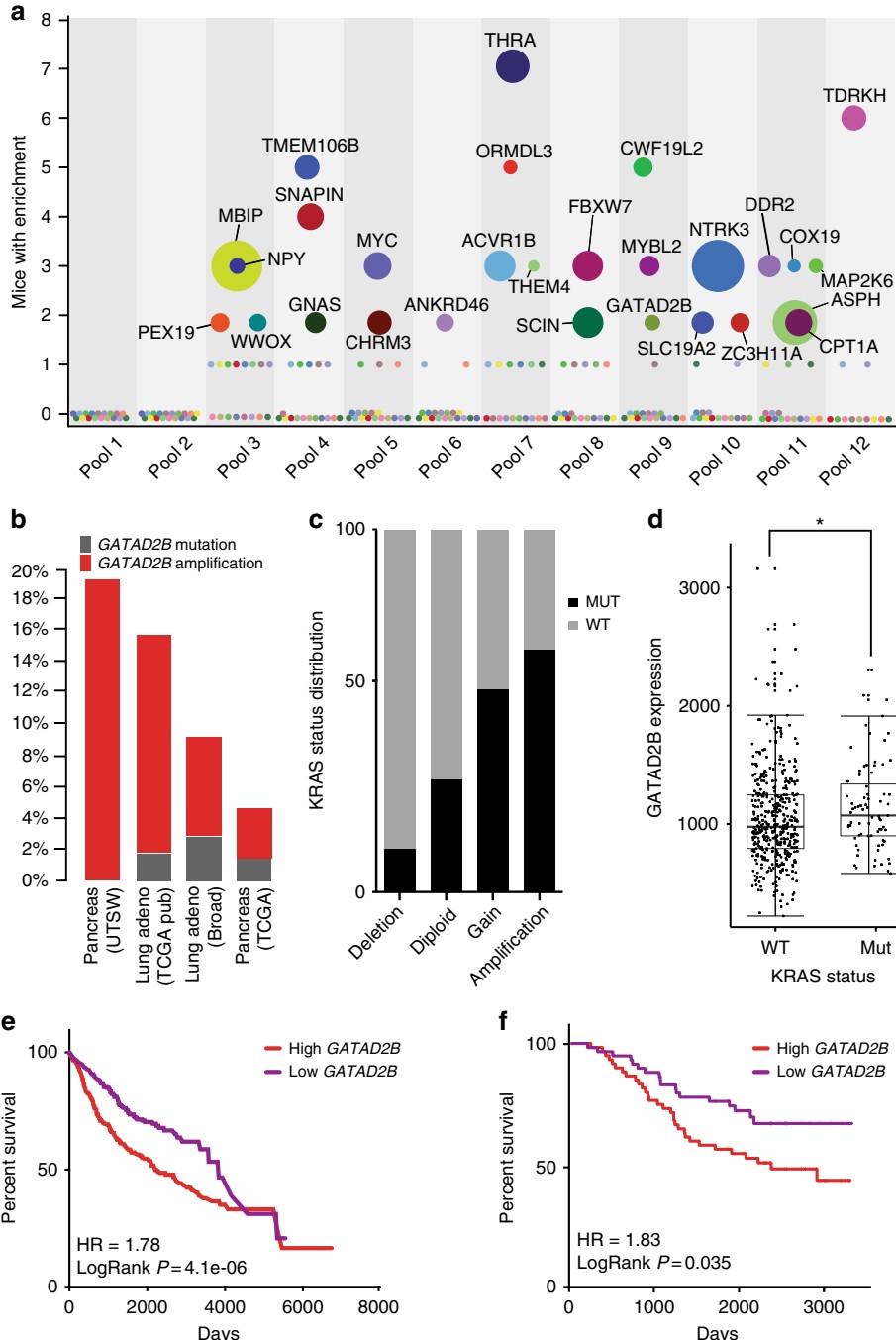

**Fig. 2** In vivo metastasis driver screening identifies *GATAD2B*. **a** Barcode sequencing analysis identifies 28 gene candidates enriched in lung metastases present in at least two mice per pool cohort. Individual genes are color-coded. Circumference of circle correlates with average enrichment in lung tissue vs. input (see Supplementary Data 1). **b** *GATAD2B* amplification and mutation across *KRAS*-driven cancers as reported by TCGA and others[4,9,29]. **c** Frequency of *GATAD2B* putative GISTIC copy number status in *KRAS* mutant ($n = 75$) vs. WT ($n = 437$) patients, (chi-square statistic 46.1108 3DF, *p-value < 0.00001. **d** Expression levels of *GATAD2B* in *KRAS* mutant (RNAseq expression 980.8) vs. wild type (1072.0) lung adenocarcinoma patients ($n = 517$, Wilcoxon rank-sum test, $p < 0.0350$). **e, f** Increased *GATAD2B* expression correlates with worse patient prognosis in (**e**) lung adenocarcinoma patients[32] and (**f**) patients with *KRAS*-driven lung adenocarcinomas[34]

*GATAD2B* copy number status in *KRAS* mutant ($n = 75$) vs. WT ($n = 437$) patients, we identified a significant distribution of *GATAD2B* putative GISTIC copy number gain and amplification in *KRAS* mutant compared to wild-type, (chi-square 46.1108, 3 DF, $p < 0.00001$; Fig. 2c). Finally, consistent with the previous NSCLC dataset (Supplementary Table 2), *GATAD2B* copy-number driven expression (****$p < 0.0001$, *t*-test, Supplementary

Fig. 3a) was higher in *KRAS* mutant compared to wild type lung adenocarcinomas ($n = 517$ patients, Wilcoxon rank-sum, $p < 0.0350$, Fig. 2d).

Survival analysis of 1145 lung cancer patients[32,33], indicated worse overall outcome for those whose tumors expressed high vs. low levels of *GATAD2B* (HR = 1.49, $P < 2.6E$-6, log rank test; Supplementary Fig. 3b), a finding that was even more significant

among patients with adenocarcinoma (HR = 1.78, $P < 4.1e-06$, log rank test; Fig. 2e). We found no correlation between *GATAD2B* expression and outcome among patients with squamous cell carcinoma, a lung cancer subtype rarely associated with mutationally activated *KRAS* (HR = 1; Supplementary Fig. 3c)[32]. Importantly, elevated *GATAD2B* expression correlated with poor patient survival specifically in *KRAS*-driven lung cancer as determined by Rousseaux et al., further suggesting that *GATAD2B* may enhance *KRAS*-driven tumor growth in lung adenocarcinoma patients (HR = 1.83, $P = 0.035$, log rank test; Fig. 2f)[33,34].

**GATAD2B promotes pro-tumorigenic and pro-metastatic activity**. To functionally examine the effect of increased *GATAD2B* expression in the context of activated *KRAS*, we constructed a primary lung cell model to permit regulated expression of *KRAS^{G12D}*. To do this, we leveraged a non-transformed, immortalized, human primary bronchial epithelial cell line (HBEC; *hTert*, *CDK4*, *TP53* knockdown) that remains anchorage dependent and does not develop tumors when implanted into mice[35]. We next modified these cells by stably integrating a regulatable *KRAS^{G12D}* allele, *iKRAS^{G12D}*[11], such that expression of mutant *KRAS* at physiological levels is activated upon addition of doxycycline (Dox 500 ng/ml; Fig. 3a). RNA profiling of HBEC-*iKRAS^{G12D}* and HBEC-*iKRAS^{WT}* cells revealed widespread changes in gene expression for HBEC cells harboring the activated *KRAS^{G12D}* allele in the presence of Dox when compared to the wild-type *KRAS* allele (2869 gene probes; fold change > 1.5-fold for on-Dox vs. off-Dox, $p < 0.01$, *t*-test; Fig. 3b and Supplementary Data 3). Of the *KRAS^{G12D}*-induced genetic changes, the Molecular Signatures Database (MSigDB; [http://software.broadinstitute.org/gsea/msigdb/index.jsp]) identified the oncogenic *RAS* signature as a top-enriched gene set ($p = 2.41e-92$, one-sided Fisher's exact test; Supplementary Data 4). Upregulation of *RAS* signaling in Dox-treated HBEC-*iKRAS^{G12D}* cells was also supported by a significant overlap with a *KRAS* signature previously characterized by Singh et al. ($p > 2.1^{e-11}$)[36].

We next used the HBEC-*iKRAS^{G12D}* cell model to assess the effect of expressing constitutive *GATAD2B* in the presence and absence of *KRAS^{G12D}* induction by doxycycline. We first applied HBEC-*iKRAS^{G12D}* cells to in vitro Matrigel cell invasion assays, which indicated that induction of *KRAS^{G12D}* alone led to a modest increase in invasive capacity (Fig. 3c; $p < 0.0226$, *t*-test) compared to Off-Dox control cells. Stable expression of *GATAD2B* had no effect on the ability of HBEC-*iKRAS^{G12D}* cells to invade matrix in the absence of Dox; however, simultaneous induction of *KRAS^{G12D}* expression in the setting of constitutive *GATAD2B* over-expression led to a synergistic effect on cell invasion (\*\*$p < 0.01$, 2 way- ANOVA; Fig. 3c). The observed increase in cell invasion in the 22 h assay was not due to an increase in *GATAD2B*-mediated cell proliferation, as proliferation assays indicated no difference between GFP-expressing and *GATAD2B*-expressing cells in the presence of Dox over a 4-day period (Supplementary Fig. 4a), however; we did observe a marked decrease in proliferation upon addition of *KRAS^{G12D}*, indicating an oncogene-induced senescent phenotype in these non-transformed cell lines, which was confirmed by staining for beta-galactosidase (Supplementary Fig. 4b).

We next examined whether *GATAD2B* expression would influence HBEC-*iKRAS^{G12D}* tumor growth and/or metastasis when subcutaneously implanted into the flanks of athymic mice. GFP control-expressing HBEC-*iKRAS^{G12D}* cells did not form tumors in mice maintained off Dox chow (Fig. 3d, orange line, $N = 10$), and randomizing mice injected with these cells onto a Dox diet led to small palpable tumors (average ~200 mm³) that did not progress

beyond their small size over the ~6 month duration of the tumor assay (Fig. 3d, blue line, $N = 10$). Similarly, HBEC-*iKRAS^{G12D}* stably expressing *GATAD2B* did not develop tumors in the absence of Dox (black line, $N = 4$); however, mice injected with these same cells developed robustly growing tumors when fed a Dox diet to turn on *KRAS^{G12D}* expression (red, $N = 8$ and purple lines $N = 2$). To determine whether *GATAD2B*-expressing HBEC-*iKRAS^{G12D}* tumors (on Dox) require continued *KRAS^{G12D}* expression for tumor maintenance, a subset of tumor-bearing mice were transitioned to an Off-Dox diet (On > Off, purple line). These tumors regressed to baseline (compared to HBEC-*iKRAS^{G12D}*-GFP on Dox) thus indicating tumor growth dependence on activated *KRAS*, and transitioning animals back onto a Dox diet led to tumor re-development (Fig. 3d, purple line). Similarly, the non-progressive GFP control-expressing HBEC-*iKRAS^{G12D}* tumors that developed on doxycycline also regressed when Dox was removed (green line, $N = 6$). Because *GATAD2B* has a known role in chromatin modification and transcriptional control,[37–39] we verified that *GATAD2B* expression neither enhanced *KRAS^{G12D}* expression from the inducible promoter nor influenced MAPK signaling as assessed through immunoblot analysis of phospho-ERK (T202/Y204) in the presence or absence of Dox (Supplementary Fig. 4c-d). Taken overall, these in vivo data demonstrate that *KRAS^{G12D}* expression is necessary for tumor formation, but insufficient to support progressive tumor growth, while combined expression with *GATAD2B* drives progressive tumor growth that is dependent on continued *KRAS^{G12D}* signaling.

Immunohistochemical analysis of HBEC-*iKRAS^{G12D}*-*GATAD2B* SQ tumors (On Dox) verified increased levels of *GATAD2B* expression, which was consistent with RT-qPCR analysis of the *GATAD2B* transcript compared to GFP-expressing tumors (Supplementary Fig. 4e), and proliferation as determined through staining for KI-67 (Fig. 3e). Consistent with the ability of *GATAD2B* to drive in vivo metastasis of 393 P cells in our primary screen, necropsies of tumor-bearing HBEC-*iKRAS^{G12D}*-*GATAD2B* mice revealed distant metastases ($N = 4/5$ mice examined) to multiple lymph nodes and other organ sites (Supplementary Fig. 4f). Isolation of genomic DNA from primary tumor followed by barcode sequencing revealed robust enrichment of *GATAD2B*-associated barcode as expected for SQ tumor site (Supplementary Fig. 4g) and metastasis-infiltrated tissues originating from *GATAD2B*-transduced HBEC-*iKRAS^{G12D}* cells (Fig. 3f). In contrast, multiple control tissues analyzed from mice bearing HBEC-*iKRAS^{G12D}*-GFP primary tumors exhibited complete absence of GFP-associated barcode, which is consistent with the inability of HBEC-*iKRAS^{G12D}* cells to metastasize.

We sought to determine whether *GATAD2B* is required for oncogenic growth of *KRAS*-mutant human NSCLC. To do this, we first validated the efficacy of two independent shRNA hairpins (sh-3 and sh-5) targeting *GATAD2B* compared to non-targeting vector (shNT) (Supplementary Fig. 5). We next assessed the effect of *GATAD2B* depletion on anchorage-independent colony growth by five NSCLC cell lines with and without mutant *KRAS*: NCI-H23, A549, CALU1, NCI-H1437, and NCI-H1568. In all three *KRAS* mutant lines (H23, A549, CALU1), depletion of *GATAD2B* using both *GATAD2B* shRNAs robustly attenuated colony formation compared to shNT control, while no significant differences were observed in *KRAS* wild-type lines (H1437 and H1568) (Fig. 3g). Taken together, these data support the role of *GATAD2B* as a *KRAS*-dependent driver of oncogenesis and metastasis.

**GATAD2B enhances tumor growth in vivo**. While culture-based studies using engineered cell models such as HBEC-*iKRAS^{G12D}* are genetically tractable, they fail to recapitulate the microenvironment critical for the growth of human cancer cells.

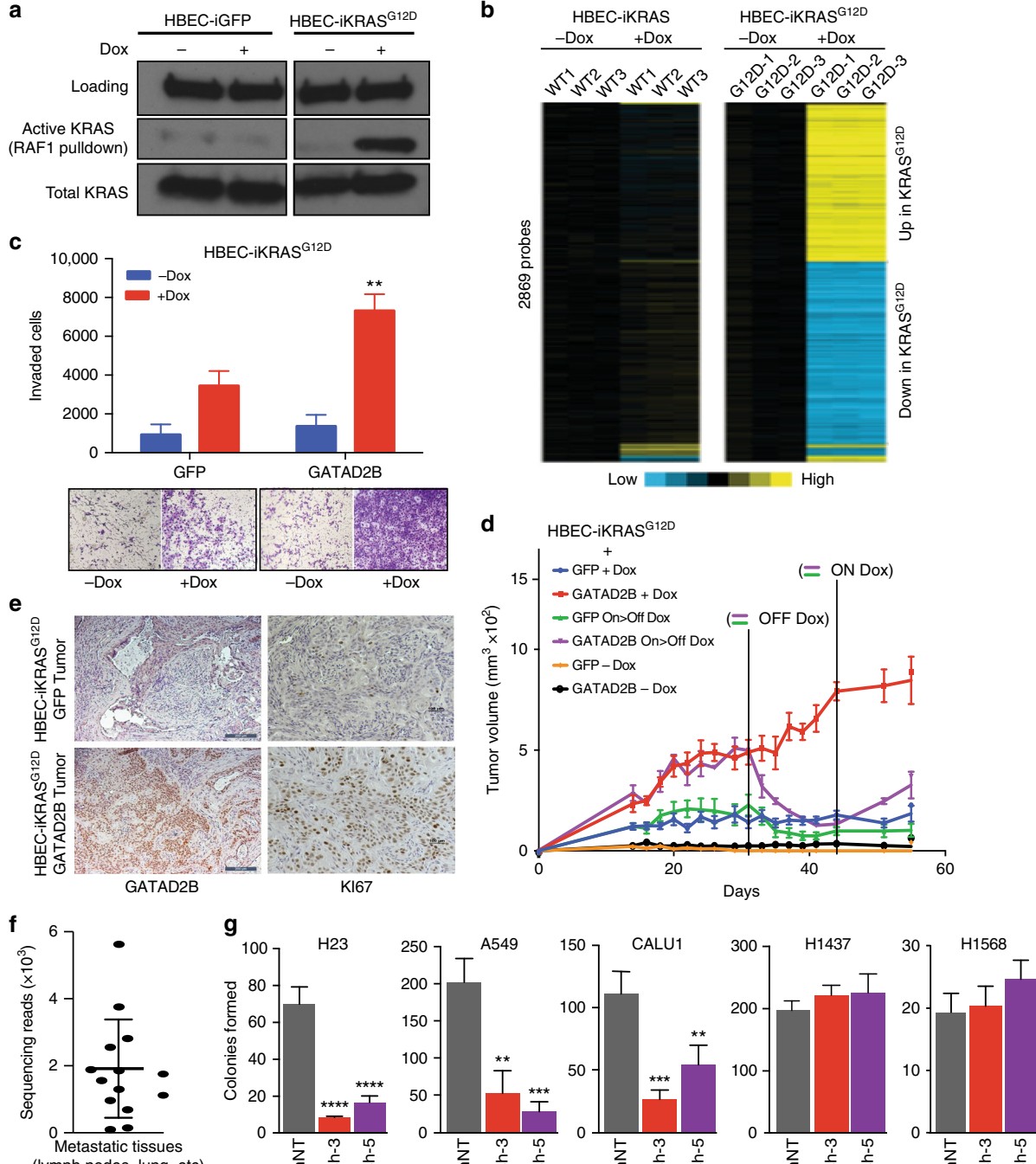

**Fig. 3** *GATAD2B* drives *KRAS*-dependent HBEC tumor growth and metastasis. **a** Immunoblot analysis of total and activated KRAS (via RAF1 pulldown assay) extracted from HBEC cells expressing GFP or *KRAS*[G12D] in the presence or absence of Dox. **b** Differentially expressed mRNAs in HBECs engineered with Dox-inducible alleles encoding wild-type (WT) or mutationally activated (G12D) *KRAS* ($n = 3$ each) in the presence or absence of Dox. **c** Transwell invasion assay of HBEC-*iKRAS*[G12D] cells stably expressing GFP or *GATAD2B* in the presence or absence of Dox (**$p < 0.01$, 2 way- ANOVA; $n = 3$ each). Representative images of transwell chambers shown in bottom panel. **d** Tumor growth curves illustrating the failure of HBEC-*iKRAS*[G12D] tumor formation in the absence of *KRAS*[G12D] induction [Off Dox; GFP (Orange), *GATAD2B* (Black)], rapid tumor formation in *GATAD2B* (Red) vs. GFP control (Blue) in the presence of Dox, and the ability to toggle *KRAS* expression from On to Off (On > Off) to demonstrate requirement of *KRAS* for tumor maintenance in *GATAD2B* (Purple) vs. GFP control (Green) tumors. **e** Confirmation of human *GATAD2B* overexpression (left, 200 μm scale bar) and enhanced KI67 staining (right, 100 μm scale bar) via immunohistochemical analysis of HBEC-*iKRAS*[G12D] SQ tumors expressing *GATAD2B* vs. GFP from **d**. **f** Number of barcode sequencing reads from metastatic tissues (lymph node, lung) isolated from mice harboring HBEC-*iKRAS*[G12D]-*GATAD2B* SQ tumors (On Dox) from **d**. NGS barcode sequencing was performed to quantitate presence of *GATAD2B*-specific barcode. Note that no metastases were observed in mice harboring *iKRAS*[G12D]-GATAD2B SQ tumors, thus barcode sequencing was not performed. **g** Anchorage-independent colony formation assays for *KRAS* mutant (H23, A549, CALU-1) and *KRAS* wild-type (H1437, H1568) NSCLC cells ($N = 3$). shNT = Non-targeting control

Therefore, we leveraged a well-established GEM model of lung adenocarcinoma where tumor initiation is achieved by Cre recombinase-mediated activation of a KRAS[G12D] allele (hereafter LSL-KRAS[G12D];[35]. In addition to the use of lung-specific Cre driver alleles engineered into mice[5–7,40], lung tumorigenesis in the LSL-KRAS[G12D] GEM models can be achieved through application of adeno- or lenti-Cre virus through nasal inhalation or intubation[41]. To build on previous studies that have used this approach for the co-delivery of cDNAs or other genetic elements[35,42], we devised a new lentiviral expression construct compatible with HiTMMoB that permits co-expression of bar-coded ORFs with Cre recombinase (Fig. 4a). By crossing the LSL-KRAS[G12D] mice with a strain carrying a Cre-inducible Luciferase allele [LSL-Luciferase;[43]], we generated a mouse strain such that delivery of Cre recombinase permits co-expression of KRAS[G12D] and Luciferase in the same cells (Fig. 4a). To test our lentiviral vector, we recombined ORFs encoding GFP and dominant-negative TP53 (TP53[R270H]) into separate lentiviral constructs with unique barcodes, followed by virus titering using HEK293-loxP-GFP-RFP Cre reporter cells. Intubation-mediated delivery of high titer virus (500,000 particles) encoding barcoded GFP- and TP53[R270H]-Cre to LSL-KRAS[G12D];LSL-Luciferase animals led to formation of lung tumors by 3 months as visualized by bioluminescent imaging (Fig. 4b). Animal necropsies confirmed numerous focal lesions throughout the lungs of animals intubated with either GFP-Cre or TP53[R270H]-Cre (Fig. 4c). In contrast to mice exposed to GFP-Cre, those intubated with TP53[R270H]-Cre exhibited gross metastases to proximal and distant lymph nodes (Fig. 4c) consistent with other reports demonstrating metastatic activity of the TP53[R270H] allele in the LSL-KRAS[G12D] model[44]. As expected, immunoblot analysis confirmed increased expression of TP53 in tumor-infiltrated lung and lymph node tissues from mice intubated with TP53[R270H]-Cre vs. mice who received GFP-Cre (Fig. 4e). NGS analysis of these tissues confirmed robust enrichment for TP53[R270H]-associated barcode consistent with the presence of TP53[R270H]-Cre transduced cells (Fig. 4f).

We next used this strategy to determine if elevated expression of GATAD2B would enhance tumorigenesis in the LSL-KRAS[G12D];LSL-Luciferase GEM model. Intubating animals with a reduced viral dose (50,000 particles) to achieve < 15% infection efficiency (based on optimization studies, see Methods) extended the overall latency of GFP-Cre infected mice as reported[41] and resulted in weak Luciferase positivity up to 12 months post-infection, as assessed by serial bioluminescence imaging (Supplementary Fig. 6). In contrast, delivery of GATAD2B-Cre virus at the same dosage led to robust Luciferase signal at approximately 6 months (Fig. 4g). Animal necropsies confirmed marked tumor burden in GATAD2B-Cre infected mice (Fig. 4h), which was consistent with Luciferase imaging. Immunohistochemical analysis of lung tissue from tumor-bearing mice intubated with GATAD2B-Cre virus confirmed strong GATAD2B expression and Ki-67 positive nuclei (Fig. 4h), and immunoblot analysis confirmed elevated levels of GATAD2B protein in the lungs of mice intubated with GATAD2B-Cre virus compared to GFP-Cre control virus (Fig. 4i). These data corroborate our results using the HBEC model, indicating that expression of GATAD2B in the context of activated KRAS promotes accelerated lung cancer progression.

**GATAD2B cooperates with MYC to promote tumor progression.** We next sought to identify the molecular mechanism by which GATAD2B exerts its pro-oncogenic and metastatic activities. For this we performed OncoPPi screening to map potential oncogenic GATAD2B interactors using a lung-cancer associated gene library[45], and a highly efficient protein-protein interaction (PPI) detection platform derived from BRET[n] technology[46,47].

The Nanoluc-tagged GATAD2B, serving as BRET[n] donor, was scanned across 83 lung-cancer associated genes fused with Venus-tag[45], serving as BRET[n] acceptor, in a multiple DNA titration combination fashion to generate BRET[n] saturation curve. The stringent proximity requirement (<10 nm) for efficient energy transfer from donor to acceptor allows the detection of protein partners to which GATAD2B directly binds. Statistical analysis of BRET signal, protein expression signal and BRETn saturation curves were performed to define positive PPIs with fold-over control of the area under curve ($FOC_{AUC}$) > 1.0, $p < 0.01$ $t$-test. The GATAD2B-MYC pair showed the strongest BRET[n] signal ($FOC_{AUC} > 8$, $p < 0.001$, $t$-test) and typical BRET[n] saturation curve with protein expression dependent increase of BRET signal (Fig. 5a). We further confirmed the interaction between GATAD2B and MYC with conventional GST-pull down. Indeed, Venus-flag-MYC was detected in complex with GST-GATAD2B (Fig. 5b). The observation of the GATAD2B/MYC PPI strongly suggests a potential role for MYC-driven oncogenicity in GATAD2B-driven metastatic lung-cancer with KRAS mutations.

Increased MYC activity has been previously identified across multiple cancer types, including KRAS[G12D]-driven tumors where MYC has been shown to enhance disease progression[28,48]. Interestingly, MYC was among the 28 ORFs found positively enriched in metastases isolated from our primary screen (Fig. 2a), thereby further supporting its role in lung cancer metastasis. We hypothesized that the physical association of GATAD2B with MYC could modulate expression of MYC-regulated transcripts, thus we examined the effect of GATAD2B expression on the activity of a MYC reporter construct containing three (GCCCACGTGGCCACGTGGCCACGTGGC) wild type or mutated (GCCCTCGAGGCCTCGAGGCCTCGAGGC) MYC-binding E-box elements upstream of a Luciferase coding sequence. Co-transfection of HEK293T cells with GATAD2B and the MYC reporter construct led to a 10-fold increase of Luciferase activity compared to vector control ($p < 0.001$ $t$-test); (Fig. 5c). Importantly, this GATAD2B-mediated increase in reporter activity was attenuated upon mutation of the E-box binding elements (Fig. 5c), indicating that GATAD2B activity requires the ability of MYC to bind to its target. Our hypothesis that GATAD2B modulates expression of MYC target genes is further supported by transcriptome analysis of the HBEC-iKRAS[G12D] cells stably transduced with GATAD2B, where induction of KRAS[G12D] with Dox led to significant transcriptional modulation of hallmark MYC-regulated targets (SRSF1, HSPD1, HSPE1) as determined by MSigDB ($p > 5.88e-12$, FDR 7.35e-11, Supplementary Table 3, Fig. 5d). Similarly, analysis of the HBEC-iKRAS[G12D];GATAD2B tumor protein lysates by Reverse Phase Proteomics Array (RPPA) compared to small, arrested GFP tumors showed elevated protein levels consistent with three previously published reports of MYC target pathway activation confirmed by chromatin immunoprecipitation[49], ChIP on chip of cell lines identifying high-affinity MYC-binding[50], or identified from the Pathway Interaction Database[51] (Supplementary Table 3, Supplementary Fig. 7, Fig. 5e).

To examine whether MYC is required for GATAD2B-driven tumor growth in the context of Dox-regulated KRAS[G12D] activation as in Fig. 3d, we used validated shRNA hairpin to knockdown MYC (Supplementary Fig. 8) in the HBEC-iKRAS cells overexpressing GATAD2B. Compared to mice with subcutaneously implanted HBEC-iKRAS cells treated with non-targeting shRNA and maintained on a Dox diet, MYC knockdown significantly decreased tumor formation and eliminated visible metastases (Fig. 5f). Together, these data indicate that GATAD2B physically associates with MYC and that MYC is required for GATAD2B-dependent tumor growth in KRAS[G12D]-expressing cells.

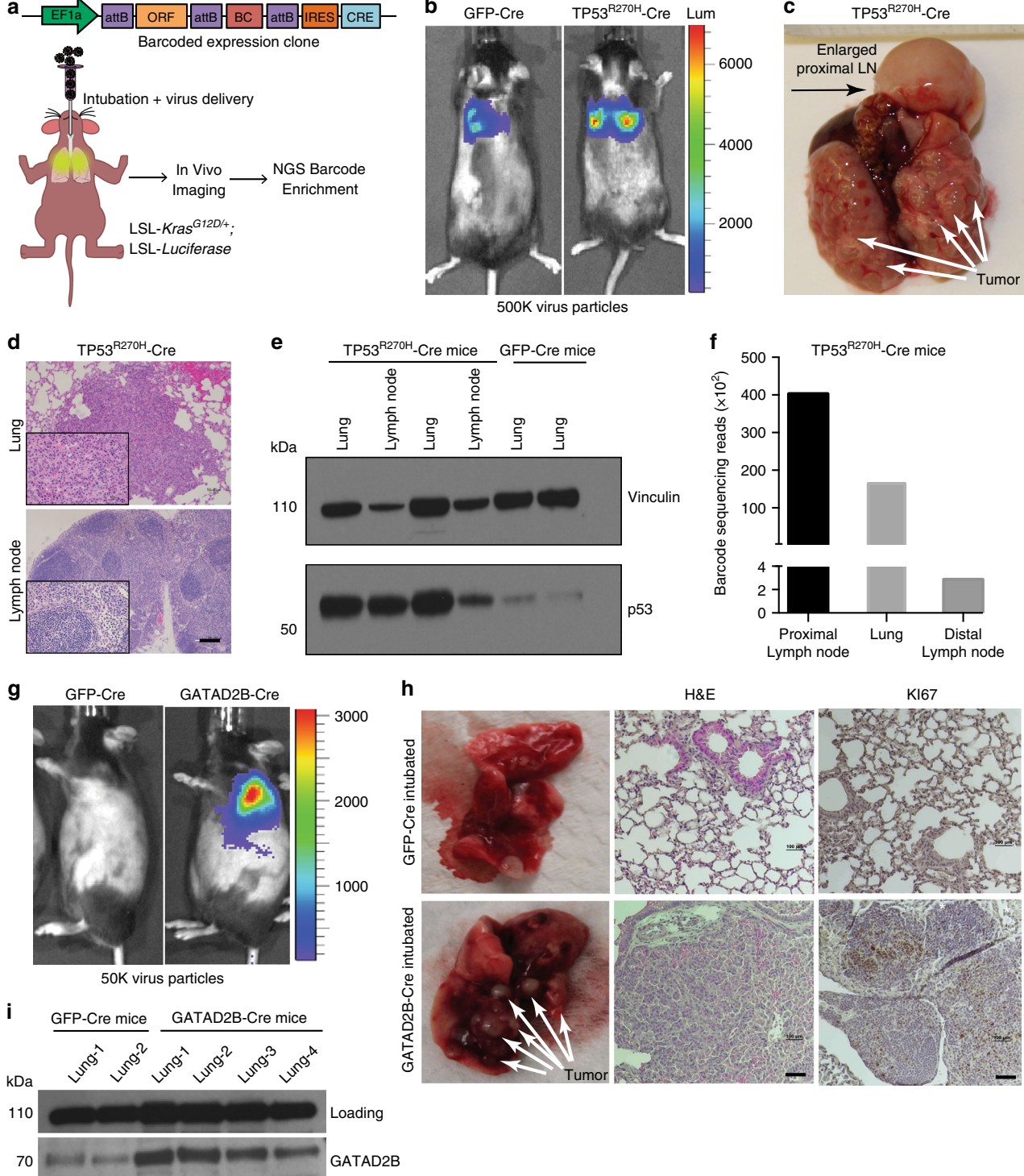

**Fig. 4** GATAD2B enhances primary lung cancer progression. **a** Schematic illustration of primary tumor and metastasis tracking with ORF-Cre lentivirus. Expression of *GATAD2B* or GFP is driven by EF1a promoter and assigned unique 24 nucleotide DNA barcode with downstream IRES element and Cre recombinase. **b** Representative Luciferase imaging of mice treated with *p53[R270H]*lentivirus (left) and GFP (right). **c** Gross morphology of representative p53[R270H]driven lung tumors and metastasis to proximal Lymph Node. **d** Representative H&E stains of resulting tumors in lung and lymph node tissue of p53[R270H] treated mice (50 μm scale bar). **e** p53 Immunoblot in lung and lymph node tissues of mice intubated with *p53[R270H]*-Cre lentivirus or GFP-Cre lentivirus. **f** Number of barcode sequencing reads from metastatic tissues (lymph node, lung) isolated from *p53[R270H]* mice. **g** Representative Luciferase imaging of mice treated with *GATAD2B* lentivirus (left) and GFP (right). **h** Representative gross morphology, H&E stains, and positive nuclei stains of the Ki-67 proliferation marker of resulting tumors in *GATAD2B* treated mice (bottom) vs. GFP (top). **i** Immunoblotting for GATAD2B in lung tissues in mice intubated with *GATAD2B*-Cre lentivirus or GFP-Cre lentivirus

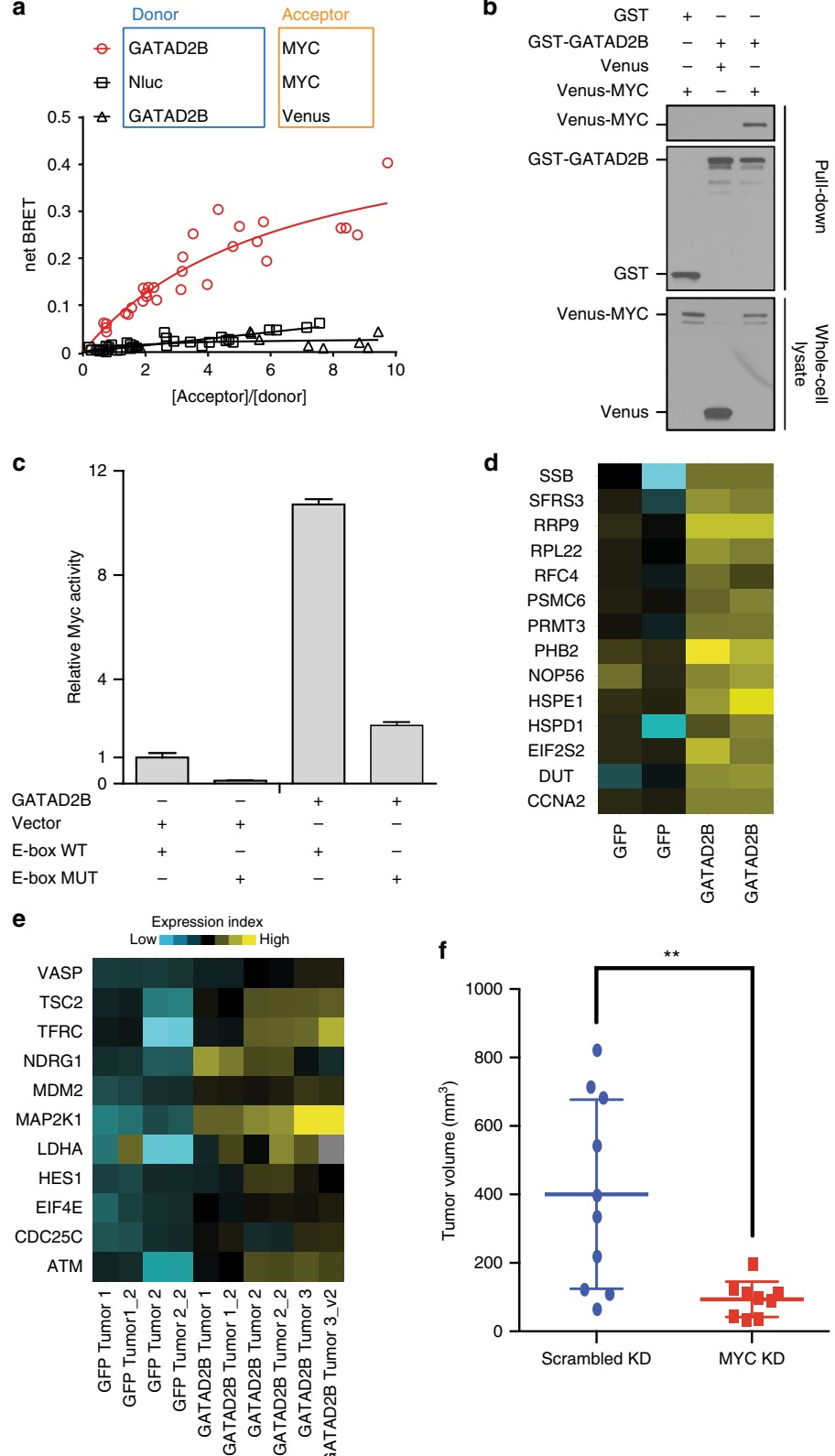

**Fig. 5** GATAD2B interacts with C-MYC to enhance KRAS driven tumor growth. **a** BRET$^n$ saturation curve of GATAD2B-MYC interaction identified from OncoPPi screen. **b** GST-pull down validation of GATAD2B-MYC interaction. **c** MYC-reporter assay revealing the GATAD2B-induced increased MYC transcriptional activity. Mutated E-box was used as negative control. **d** Differentially expressed MYC-regulated mRNAs (hallmark MYC targets, MSigDB, $p < 5.88E-12$, FDR $q < 7.35E-11$) in *GATAD2B* vs GFP *KRAS$^{G12D}$* activated HBEC cells. **e** Differentially expressed proteins of direct MYC-target genes in *GATAD2B-* ($N = 6$) and GFP-driven tumors ($N = 4$). **f** End point tumor volume analysis (**$p < .0057$, Mann–Whitney U) of GATAD2B-overexpressing HBEC line with MYC knockdown ($n = 9$) compared to scrambled control ($n = 10$)

## Discussion

Metastatic disease remains an elusive target for cancer treatment and remains the primary event defining outcome during the course of treatment for lung cancer patients. Despite major improvements made with early detection, and increased knowledge and identification of actionable tumor biomarkers, *KRAS*-driven lung adenocarcinoma still represents a commonly diagnosed form of NSCLC for which there are no targeted agents. Therefore, the identification of genes that work in concert with *KRAS* to drive tumor progression and/or metastasis provide the greatest hope of improving patient outcomes through personalized treatment options, either by acting as a direct target or targeting converging signaling pathways.

Here we performed an in vivo functional screen of 217 genetic aberrations identified from lung cancer genomics datasets. Our primary screen generated a functionally annotated list of 28 genes that led to robust metastatic dissemination and merit further interrogation, including some that have been previously identified to be mutated in lung adenocarcinoma (*GNAS, MAPK6, ACVR1B, FBXW7, NTRK3*). While this functional screening strategy has identified genetic aberrations with in vivo metastatic ability, individual functional assessment of genes scoring as metastasis drivers requires validation with additional evidence to robustly delineate their roles in the malignant phenotypes and underlying mechanisms of action. The newly developed tools outlined here allowed us to demonstrate the context-specific transformative capabilities of *GATAD2B* and will serve for the study of other genes that may be dependent on *KRAS* co-mutation.

GATAD2B is a member of the Nucleosome Remodeling complex (NuRD), 1 of 4 major ATP dependent chromatin remodeling complexes[52,53]. The major roles of the complex are regulation of transcription, chromatin assembly, cell cycle progression, and genomic stability[27]. The NuRD complex primarily functions through recruitment of HDACs to discrete loci and has been previously implicated in tumor progression and metastasis through the complex members, metastasis-associated proteins 1 and 2 (MTA1 and MTA2)[38,54–56]. Despite these known roles of the Nucleosome Remodeling and Deacetylating (NuRD) complex in driving metastatic activity, *GATAD2B* has not been previously implicated in the metastatic capabilities of this complex. To date, GATAD2A and GATAD2B have been demonstrated to be structural components that bind to histone tails[28].

We demonstrate that GATAD2B enhances tumor growth, at least in part, through a direct interaction with MYC, a known potent oncogenic and metastasis driver that also scored in the in vivo screen. Poor prognostic outcomes for patients harboring mutant *KRAS* and *MYC* amplification have been previously reported in human cancers[9,57]. In human bronchial epithelial cell (HBEC) models, *MYC* greatly enhanced aggressiveness of *KRAS* neoplastic transformation and induction of epithelial-to-mesenchymal transition (EMT)[58,59]. In addition to human studies correlating elevated MYC levels with worse clinical outcomes, disease progression, and metastasis[48,60,61], in animal models, spontaneous metastatic spread and rapid tumor progression is observed in *KRAS* mutant tumors with exogenous MYC[62,63].

We identified activation of downstream MYC target genes in vitro and in vivo in *GATAD2B*-driven tumors. This is an intriguing finding because NURD activity has not been previously linked with physical interactions to oncogenic c-MYC, although the MTA1 subunit of the NURD complex has been shown to be a transcriptional target of c-MYC, required for Myc-induced transformation of cells[55]. This finding suggests an important level of whole-genome epigenetic reprogramming that is required for the progression and metastasis of *KRAS*-mutant lung cancer.

The identification of convergence between *KRAS* signaling and NURD/MYC epigenetic reprogramming may inform evidence-based clinical trial development for metastatic lung cancer patients with activated *KRAS* mutations, or even potentially other tumor types with *KRAS* mutations. Given that *KRAS* and *MYC* are currently therapeutically undruggable genetic alterations found in many tumor types, a greater understanding of the interaction and cooperativity between these oncogenic changes and *GATAD2B* may open up the exploration of new therapeutic approaches. Analysis of primary tumors in our study showed elevated signaling pathways commonly shared between tumor growth drivers and metastasis-inducing genes. These include HIF1A and hypoxia[64], MET pathway activation[65] and mTOR signaling[66]. Further in-depth study will be needed to understand what role these pathways play in the malignant phenotypes we observed. This study describes a functional screening platform that can readily identify novel candidates, several of which have been validated herein or in the companion paper on TMEM106B[67], and which are poised for future studies. More studies are warranted to determine whether GATAD2B may be targeted therapeutically, and given the role of GATAD2B as a NuRD complex member, emerging therapeutic options such as selective HDAC or BRD4 inhibitor based-strategies may be warranted.

## Methods

**Gene candidate selection**. We utilized data collected from GEM models with a latent, somatically activated *Kras* allele in a mutant *p53* background (*Kras^G12D^*, *p53^R172HΔG^*). Mice harboring these mutations develop metastatic lung adenocarcinoma[68], the most common histological subtype of non-small cell lung cancer (NSCLC), and the combination of *Kras^G12D^* and mutant *p53* is frequently found in human NSCLC[24]. This GEM model was used to derive a panel of lung tumor cell lines confirmed to be heterozygous for both the *p53^R172HΔG^* and *Kras^G12D^* alleles, and detailed characterization of these lines confirmed molecular and physiological properties that mirror human NSCLC[18]. Gene expression patterns of the *p53^R172HΔG^*, *Kras^G12D^* tumor explants using the metastasis-incompetent 393 P cell line as reference revealed 3000 genes were differentially expressed (fold-change increase or decrease > 1.5; *p* < 0.01, paired *t*-test) in metastasis capable 393LN and 344SQ tumors[17]. Overlapping genes found up-regulated in both comparisons were next triangulated with human copy number amplifications documented by TCGA (1.5-fold somatic amplification across ≥ 5% of 154 analyzed lung adenocarcinoma specimens). Additional genes with identified hyperactivating mutations were integrated[24] that focused on 623 genes with potential relationships to cancer. Filtering copy number amplifications from TCGA with the 1013 non-synonymous somatic mutations identified by their study revealed 31 amplified genes that contain at least one validated missense mutation within the 188 analyzed tumors.

**Cell culture**. All cell lines were propagated at 37 °C and 5% $CO_2$ in humidified atmosphere. 393 P cells were published previously[18] and were cultured in RPMI media supplemented with 10% Fetal Bovine Serum, 1%Penicillin/Streptomycin. HBEC cells were a gift from J. Minna (University of Texas-Southwestern Medical center, see[35] and were cultured in KSFM medium (Life Technologies). NSCLC cell lines were purchased from ATCC and cultured following their recommendations. In some experiments as indicated, cells were propagated in Doxycycline (500 ng/ml). The Dox-regulatable *KRAS^G12D^* HBEC cell line was constructed using the pInducer system[69] and constructed as described for HPDE-i*KRAS^G12D^* cells previously[11].

**In vivo screen and barcode enrichment analysis**. For in vivo screening, pools of 20 cell lines were prepared with 50,000 cells per virally infected line containing 19 cell lines expressing individual ORFs and 1 expressing an mCherry control. These cells were mixed and re-suspended in RPMI solution without serum and Matrigel (BD Bioscience) for SQ implantation into 129 Sv at 1 million cells per site.

**Barcode sequencing**. gDNA was extracted from library-infected 393 P cells (injected or Input) and individual tumor cores (output) for quadruplicate and triplicate, respectively, as done previously[11]. Barcode enrichment was assessed by quantitating the number of occurrences for each barcode sequence as a ratio to total number of barcode reads in each sample. Standard deviations were calculated for duplicate and triplicate reactions (where available) for input and output samples.

**NSCLC cell line assays**. Soft agar assays were performed in 6-well plates in triplicate. First, bottom layers were prepared at 0.8% Noble agar (Affymetrix, Inc.) with complete RPMI growth medium. After solidification, 1000 cells were mixed with 0.45% agar in complete growth medium and laid on top of the bottom layer. Two milliliters medium was added in each well after 3 days and medium was refreshed every 3 days. Colonies were counted 4–5 weeks after seeding. For knockdown studies, *GATAD2B* pLKO shRNAs were purchased from Thermo (TRCN0000230881, TRCN0000015315, TRCN0000230880), MYC shRNA (V2LHS_152051, Dharmacon) and used to transduce cells following manufacture's recommendations. Cell invasion assays were done as previously described[25]. For cell proliferation assays, transduced HBECs cells were pretreated with Dox (500 ng/ml) for 48 h, followed by plating 1000 cells onto White Opaque 96-well microplates in quadruplicate, respectively. Cell proliferation was assayed by CellTiter-Glo® (Promega) using a Wallac Victor2 Multilabel Counter at multiple time intervals, according to manufacturer instructions and previously[11]. Cells were stained for senescence-associated β-galactosidase activity according to manufacturer's instructions (Cell Signaling Technology). All data were assessed by two-tailed *t*-test calculation using Prism 6, error bars represent s.e.m. (Graphpad).

**Animal studies**. All studies using mice were performed in accordance with our IACUC-approved animal protocols at Baylor College of Medicine and The University of Texas at MD Anderson Cancer Center. For xenograft tumor assays, $10^5$ cells were re-suspended in a 1:1 Hank's balanced salt solution (Life Technologies) and Matrigel (BD Biosciences) and injected into female athymic mice subcutaneously at bilateral flanks. Mice were monitored twice a week and tumors were measured and calculated by length × width$^2$ / 2. Lung intubation studies were performed as described previously[41]. Lentivirus was produced using standard virus packaging vectors and virus protocols. Virus was filtered with 0.45 μm filter followed by ultracentrifugation at 25,000 RPM for 3 h and resuspended in Hank's Balanced Salt Solution (HBSS). Cells were titered by transducing a Cre reporter cell line (amsbio SC018-Bsd) with ~ 15% transduction efficiency (for optimal MOI = 1). Subsequent luciferase imaging was completed with D-Luciferin (Gold Biotechnology) using the IVIS Lumina II. Xenograft studies for MYC knockdown experiment were performed as described earlier, pGIPZ MYC shRNAs (Clone ID: V2LHS_152051, Dharmacon) prepared following manufacture's recommendations cells were suspended in HBSS at $2.4 \times 10^5$/ injection and combined in a 1:1 ratio with matrigel matrix (BD).

**Immunoblotting and quantitative RT-PCR**. Cells and tumor tissues were lysed with RIPA buffer with protease inhibitors cocktail (Sigma Aldrich) and phosphatase inhibitors cocktail (Calbiochem). Protein lysates were separated on 4–12% Bis-Tris gel (Life Technologies) and transferred to PVDF membranes. The following antibodies were used to detect protein expression: GATAD2B (Sigma Aldrich 1:1000), Vinculin (Cell Signaling Technologies 1:1000), Erk1/2 (Cell Signaling Technologies 1:1000) Vinculin (CST, 1:10000) and GAPDH (Santa Cruz 1:25,000). Uncropped blots are available in the supplementary files.

**Immunohistochemistry**. Murine lung tissues or subcutaneous tumors were excised, washed in PBS, and fixed in formalin for 48 h. Once fixed, samples were dehydrated in graded ethanol series followed by xylene. Samples were then embedded in paraffin, sectioned onto slides (5 mm thick), and allowed to dry. For H&E staining, slides were deparaffinized using standard procedure (Xylene × 2 washes, 100% ethanol × 2, 95% ethanol, 70% ethanol, 50% ethanol), and stained with H&E (Vector H-3401; Sigma HT110132). For immunohistochemical staining, slides were processed using Vectastain Elite ABC HRP Kit and DAB Peroxidase Substrate Kit (Vector Laboratories, PK-6101, SK-4100). Briefly, after deparaffinizing procedure (as per above), antigen retrieval was completed using 0.01 M sodium citrate buffer (pH 6.0), for 15 min at 95 C, followed by blocking with 0.3% H2O2 for 30 min. Slides were stained for GATAD2B antibody (1:5000, Sigma, HPA-017015) or Ki-67 antibody (1:500, Cell Signaling, 9027, hum; 1:500, Abcam, ab16667, mus) and processing and staining were completed using Vectastain and DAB kits in accordance with instructions provided by the manufacturer. qPCR primers were used as the following: (GATAD2B: F-5′-CAAAAGCTGTGCCTCA CTTC R-5′-TTCCAGTGAGGGGTGAAATC KRAS: F-5′-GGACTGGGGAGGC TTTCT R-5′-GCCTGTTTTGTGTCTACTGTTCT MYC: F-5′- CATCCTGTCCG TCCAAGCAG R-5′- CCTTACTTTTCCTTACGCACAAGA) qPCR was performed as previously described[25].

**GATAD2B interactors screening**. A BRET$^n$ technology-based OncoPPi screening platform was used to scan GATAD2B across lung-cancer associated gene library[45]. Briefly, NLuc-tagged GATAD2B was co-transfected with Venus-tagged gene in pairwise into HEK293T cells seeded in 1536-well plate. Two sets of 1536-well plates, one white plate (Corning) for BRET signal measurement and another black plate (Corning) for the measurement of Venus fluorescence intensity, were prepared side-by-side as previously described[46,47]. Linear polyethylenimines (Polysciences) were used as transfection reagent throughout the study. 384-cannula array integrated with Sciclone ALH 3000 liquid handler (PerkinElmer) was used to assist high-throughput transfection. 48 h after transfection, Nano-Glo® luciferase

substrate furimazine was added to the cells directly. PPI signal, NLuc-GATAD2B expression and Venus-genes expression were monitored by measuring BRET ratio, luminescence signal and fluorescence signal, respectively, using an Envision Multilabel plate reader (PerkinElmer). The BRET ratio is calculated as the ratio of intensity at 535 nm over 460 nm. The net BRET is the bleed-through corrected BRET ratio. The relative amount of NLuc-GATAD2B and Venus-gene partners expressions were measured by the luminescence signal at 460 nm ($L_{460}$) and the laser-excited Venus fluorescence intensity (FI). The ratio of relative amount of acceptor over donor protein expression (Acceptor/Donor) was defined as Venus FI/$L_{460}$. Area under the BRET saturation curve (AUC) was built by fitting the net BRET on *Y*-axis and Acceptor/Donor on *X*-axis using one-site binding equation, and the statistical significance was calculated with the unpaired two-tailed *t*-test as described previously[46,47].

**MYC reporter assay**. MYC-luciferase reporter assay was conducted essentially as described previously[45,70]. HEK293T cells were transfected with Venus-Flag-GATAD2B or Flag-Venus vector along with the Myc-E-box-containing Firefly luciferase reporter plasmid. Renilla luciferase expression vector served as a calibration control. The proteins were expressed in cells for 48 h in DMEM media supplemented with 10% FBS. Then, cells were harvested, and transferred to a 384-well plate (20 μL per well). MYC reporter assay was performed using Dual-Glo luciferase kit (Promega) following manufacturer's instructions. The normalized luminescence was calculated as a ratio of luminescence of Firefly luciferase to the luminescence of Renilla luciferase. The statistical significance was calculated with the unpaired two-tailed *t*-test.

**Transcriptome profiling and RPPA**. RNAs extracted from the indicated cell lines cultured with or without Dox (500 ng ml for 48 h) were processed for hybridization on Illumina Human Bead Array by the Baylor College of Medicine Genome Profiling Core Facility. Array data were deposited in NCBI Gene Expression Omnibus (GEO) database (http://www.ncbi.nlm.nih.gov/geo/) under accession number GSE100336. RPPA was performed by the MD Anderson RPPA Core Facility using protein lysates extracted from the indicated subcutaneous tumors. For each RPPA replicate, cells were washed with cold PBS twice and lysed with 100 μl lysis buffer (1% Triton X-100, 50 mM HEPES, pH 7.4, 150 mM NaCl, 1.5 mM MgCl2, 1 mM EG/TA, 100 mM NaF, 10 mM Na pyrophosphate, 1 mM Na3VO4 and 10% glycerol, containing freshly added protease and phosphatase inhibitors from Roche Applied Science Cat. # 05056489001 and 04906837001, respectively), mixed with $4 \times$ SDS sample buffer (40% Glycerol, 8% SDS, 0.25 M Tris-HCl, pH 6.8) and boiled for 5 min before RPPA. Two-sided homoscedastic *t*-tests (using log-transformed data) and fold changes were used to determine differentially expressed genes and proteins. For transcriptome data, gene probes significant with fold change > 1.5 and *p* < 0.01 Dox vs. no Dox for either KRAS WT or KRAS G12D cells were clustered using a "supervised" clustering approached as previously described[71]. Expression patterns were visualized as heatmaps using JavaTreeView or R statistical software. MSigDB (version 3.0) gene sets were searched using SigTerms[72]. Grubbs test was used to remove significant outliers where appropriate.

**Data availability**. Array data were deposited in NCBI Gene Expression Omnibus (GEO) database (http://www.ncbi.nlm.nih.gov/geo/) under accession number GSE100336. All other relevant data are available from the authors.

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

## Acknowledgements

The authors would like to thank Dr. John Minna (UT Southwestern) for providing parental Human Bronchial Epithelial Cell lines. C.G. also wishes to thank Dr. Sharon Plon for her guidance and mentorship during this project and in review of this manuscript This project was supported in part by the Genomic and RNA Profiling Core at Baylor College of Medicine with funding from the NIH/NCI grant (P30CA125123) and Cytometry and Cell Sorting Core Facility with funding from the NIH (P30 AI036211, P30 CA125123, and S10 RR024574). This project was also supported by the Cancer Prevention and Research Institute of Texas (CPRIT; RP140216) by funding to K.L.S., the Department of Defense (LC110216) by funding to K..L.S. and D.LG., an MD Anderson Cancer Center Physician Scientist Award to D.L.G., and by the NIH (U01CA168394) by funding to K..L.S. and G.B.M. H. L. was supported by the CPRIT Pre-Doctoral Fellowship (RP140102). D.L.G. is a Lee Clark Fellow of the University of Texas MD Anderson Cancer Center. This project was also supported by the Cancer Target Discovery and Development Network grants U01CA168449 and U01CA217875 to H.F.

## Author contributions

Conception and design: C.G., S.T.K., D.L.G. and K.L.S. Acquisition of data: C.G., S.T.K., O.Z., X.L.M., A.A.I., H.L., R.H.C., Y.H.T., D.M., M.M., K.E., J.J.F., S.A., F.C., Z.C. Analysis and interpretation of data: C.G., S.T.K., O.Z., F.C., K.C., C.J.C., D.L.G. and K.L.S. Writing, review, and/or revision of the manuscript: C.G. S.T.K., D.L.G. and K.L.S. Technical and material support: G.B.M. Study Supervision: H.F., D.L.G. and K.L.S.

## Additional information

**Competing interests:** The authors declare no competing interests.

