## [Peer Review File · Nature Communications]

Reviewers' comments:

Reviewer #1 (Remarks to the Author):

Manuscript No: NCOMMS-17-19578

Review Comments:

The research paper by Grzeskowiak et al has a focus on functional characterization of lung cancer genome datasets to distinguish subsets of functional oncogenic driver mutations that synergize with KRAS mutant causing primary and metastatic lung adenocarcinoma. Specifically, the functional role of GATAD2B was thoroughly studied as a potent driver in this context. Overall, the content is good, novel to some part and appreciable efforts; however, following points must be addressed to improve the quality of the work -

1. In figure 1, where the intersection of different datasets have been done, comprise of genes picked from studies comparing KRAS + p53 R172H mutations bearing tumors. However it would be better to compare the unique gene signatures from only KRAS mutation harbouring tumors as well, because the gene sets derived from these datasets could be a consequence of the co-operation of p53 and KRAS mutation.
2. Supplementary figure 1, since pool 9 was the one that showed enrichment of GATAD2B (with ref from figure 2A), it would be ideal to see the data for tumor formation and lung metastasis with pool 9 injection. Secondly, under pool 3, pool 4 injected conditions, is it that only the lung nodules are seen in this case or there are other distant organs where cells can also metastasize? Explain.
3. Authors must present data (or a schematic map) for the HBEC-iKRAS G12D- GATAD2B mice showing distant metastasis to lymph node and other organ sites that they mention in result section on line 250-253.
4. Fig 3G - It would be a stronger establishment of the case to prove authors point that GATAD2B co-operates with mutant KRAS to drive tumorigenesis. This can be done in vitro, showing the effect of GATAD2B knockdown in a wild- type KRAS background cell line as for the data shown in TCGA human data analysis as well as in vivo tumor experiment as in 3D (GATAD2B off condition) brings the indication in that direction.
5. Figure 3D – to my opinion the authors have focussed on GATAD2B results to prove their point, which is fine. But on the chart, the lines that follow 'GFP on' vs. 'GFP on>off' has been ignored. The GFP result has also shown similar trend for on>off experiment, which complicate the claim that the authors are drawing here. Of course for the GFP lines the tumors are half the size than the GATAD2B lines and therefore the drop is hidden, but when one compare 'On' to 'On>off' situation for both GFP and GATAD2B, the trend in tumor size reduction over time is similar or same. Please explain with additional proof.
6. The proliferation data shows a reduced proliferation of GFP and GATAD2B cells, on induction with dox as compared to dox negative condition, which is contradictory to the hypothesis (Fig supp 4A). Perhaps, this data could be validated by a much more direct approach of measuring proliferation like EdU/BrDU assay.
7. In supplementary figure 4C, it mentioned that there is no change in pERK in presence or absence of dox, but from the immunoblot it is quite clear that there is an increase in pERK 2 (upper band), with dox treatment, in both GFP and GATAD2B cells. Whereas the pERK 1, is only increased with dox in GFP control cells and remains unaltered in GATAD2B cells. What could be the probable justification for this result?
8. Fig 4E - What is the 110 KD loading control used and why? The sample loading show so much variability? Tubulin or actin loading control should have been ideal for this case. Considering everything gone right as presented here, authors need to explain why is it that p53 expression is more in proximal lymph nodes, than the lungs?
9. Figure 4G & H – In the context of Cre and GATAD2B lentivirus experiment, as claimed in discussion (line 388-389), the role of GATAD2B in driving tumor progression and 'METASTASIS' is overstatement. These experiments only show that GATAD2B lentivirus can enhance tumorigenesis. Further, additional results and discussion is required explaining whether and how this GATAD2B-cre lentivirus create lung specific higher luciferase signal.
10. Figure 5A – What has been represented in the X-axis is not clear. BRET methodology on page

19 also do not explain how the BRET ratio is calculated – is it a simple Acceptor/donor ratio represented in Y-axis or is this bleedthrough corrected BRET ratio represented as 'Net BRET' on that axis. In any case, X-axis representation needs to be clearly mentioned. Further, what does it mean 'FOCAUC>1.0' on line 323. Authors should explain all abbreviation at the first occurrence as standard.

Reviewer #2 (Remarks to the Author):

Grzeskowiak and colleagues describe an in vivo gain of function screen for metastasis drivers in NSCLC. Starting from transcriptome profiles derived from comparing primary NSCL cell lines and corresponding metastases in combination with mouse/human cross-species comparisons, they defined a set of 251 genes potentially associated with metastasis. They used this gene list for an in vivo gain of function screen. The study then validates and functionally characterizes Gatad2d, one of the candidate discovered in the screen.

The principal findings of the study are (1) identification of novel metastasis-promoting oncogenes (28 genes emerged from this screen), (2) cross-species mouse/human comparisons to confirm the potential human relevance, (3) validation of Gatad2b as a bone-fide context-dependent (Kras mutant NSCLC) metastasis-promoting oncogene, and (4) identification of Myc as a Gatad2b interaction partner and a downstream mediator of Gatad2b-dependent effects on tumor growth and metastasis.

The screen using barcoded expression vectors is innovative and the results are persuasive. The manuscript is well written and the results are discussed in a balanced manner.

Overall this is an excellent study, which in my opinion is suitable for publication in Nature communications. It not only provides a list of metastasis drivers (which can in future be picked up by others) but also a convincing validation of Gatad2b and mechanistic insights into possible downstream effects.

Specific comments:

1. It seems that human ORFs are used in murine cancer cells (page 5, 2nd paragraph). Some ORFs might therefore not be functional in the screen (giving false negative results). Can the authors comment on this issue?
2. Please comment on the use of wild type ORFs for oncogenes altered by activating mutations in human tumours (page 5, 2nd paragraph).
3. Figure 1A: Annotation of promoter (EF1a?) in vector map is missing.
4. Supplementary Figure 4A: Kras-expressing cells (GFP + DOX and GATA + DOX) seem to proliferate markedly slower than Kras wild type cells (GFP – DOX and GATA – DOX), which is counterintuitive. Is there evidence for senescence? Please comment.

Reviewer #3 (Remarks to the Author):

This manuscript reports on a the use of in vitro and in vivo functional genetics platforms to identify mediators of lung cancer metastasis. The system and the findings are interesting and the experiments are well conducted. There are issues to address before firm conclusion can be drawn.

(1) The authors use a flank system to identify mediators of lung cancer metastasis. This system is plagued by the fact that the findings are not derived from a spontaneous lung cancer model. This needs to be experimentally addressed both for GATA2B and MYC.

(2) The mechanisms of cooperatively between KRAS and MYC is not shown. This is a major gap.

(3) The authors use KRAS mutant models but fail to show whether the axis identified is specific for KRAS biology or non-specific.

(4) The role of the signaling axis identified in tumor growth versus metastatic processes remains unclear and should be mechanistically elucidated before drawing firm conclusions.

In Vivo Screening Identifies GATAD2B as a Metastasis Driver in Kras-Driven Lung Cancer

We would like to thank the Reviewers for their insightful and constructive comments. Below we address issues brought forward and provide a detailed point-by-point response to each comment. The Reviewers' comments and resulting revisions served to greatly improve our manuscript, producing a study that we feel merits reconsideration for publication by *Nature Communications*.

Reviewer #1 (Reviewer Comments to the Author):

The research paper by Grzeskowiak et al has a focus on functional characterization of lung cancer genome datasets to distinguish subsets of functional oncogenic driver mutations that synergize with KRAS mutant causing primary and metastatic lung adenocarcinoma. Specifically, the functional role of GATAD2B was thoroughly studied as a potent driver in this context. Overall, the content is good, novel to some part and appreciable efforts; however, following points must be addressed to improve the quality of the work -

Response: We thank the Reviewer for their comments recognizing the novelty of our functional *in vivo* screening study and highlighting the comprehensive nature of our functional assessment of *GATAD2B* in the context of lung cancer. We address the Reviewer's comments below.

Comments to be addressed:

1. In figure 1, where the intersection of different datasets have been done, comprise of genes picked from studies comparing KRAS + p53 R172H mutations bearing tumors. However it would be better to compare the unique gene signatures from only KRAS mutation harbouring tumors as well, because the gene sets derived from these datasets could be a consequence of the co-operation of p53 and KRAS mutation.

Response:

We thank the reviewer for their careful consideration of our gene list, generated for large-scale, *in vivo* functional screening. The reviewer has thoughtfully raised concerns regarding the use of the *KRAS^{mut},p53^{mut}* models for the initial candidate generation analysis, with the specific concern that simultaneous expression of mutant *KRAS* and

p53 potentially limits metastasis driver gene discovery to representing those genes which are enhanced by both mutations. We fully agree with the reviewer that analysis of a *KRAS*^{mut};*p53*^{wt} dataset would extend the impact of our study. To determine whether the *p53*^{R172H} mutation filtered our *KRAS* mutant gene list further, we directly compared candidate gene expression levels between *KRAS*^{G12D};*p53*^{wt} and *KRAS*^{G12D};*p53*^{R172H} murine lung tumors obtained from our previously published transcriptome comparison of two autochthonous genetically-engineered models (PMID: 19404390). We discovered that out of all genes tested in this screen, there were only five genes (approximately 2.2% of the candidate list) with significantly higher expression in the *KRAS*^{G12D};*p53*^{R172H} dataset vs the *KRAS*^{G12D};*p53*^{wt} dataset, none of which were identified as scoring drivers in our functional assessment. Based on this comparison, we conclude that cooperation between mutant *KRAS* and *p53* in this sample set did not play a significant role in the candidate driver selection. In our revised manuscript, we have included this analysis in **Supp. Table 1**, and updated our results section accordingly.

2. Supplementary figure 1, since pool 9 was the one that showed enrichment of *GATAD2B* (with ref from figure 2A), it would be ideal to see the data for tumor formation and lung metastasis with pool 9 injection. Secondly, under pool 3, pool 4 injected conditions, is it that only the lung nodules are seen in this case or there are other distant organs where cells can also metastasize? Explain.

Response: We thank the reviewer for their excellent suggestion to include images from Pool 9, and we agree this will enhance our *GATAD2B*-centered validation efforts, therefore we have updated **Supp. Fig. 1** to include representative Pool 9 images of primary tumor and metastases as indicated below:

Response to sub-comment 2: The reviewer also raised an important question concerning the clarity with which we described the location of nodules in our primary screening system. To clarify this point, lines 156-158 discuss the location of metastases

analyzed from this model as metastasizing from a subcutaneous site to the lungs, which was a consistent finding across each of the pools containing drivers, and is consistent with our prior publications with the KP syngeneic tumor models. While we do not rule out the possibility that other sites contained metastatic lesions, observable lung metastasis events were used for streamlined primary screening. Genes identified from these lesions were subsequently and necessarily followed up with individual validation studies to investigate and corroborate the observed metastatic phenotypes. We thank the reviewer for pointing out potentially confusing language, and we have updated our rationale for this approach in the results section of our manuscript.

3. Authors must present data (or a schematic map) for the HBEC-iKRAS G12D-GATAD2B mice showing distant metastasis to lymph node and other organ sites that they mention in result section on line 250-253.

Response: We thank the reviewer for noting the absence of this important information. In addition to the barcode sequencing data in **Fig. 3F**, we have provided a schematic map as **Supp. Fig. 4F** to include locations of metastatic sites where the GATAD2B-DNA Barcode was detected in these animals and revised our results section accordingly.

4. It would be a stronger establishment of the case to prove authors point that GATAD2B co-operates with mutant KRAS to drive tumorigenesis. This can be done in vitro, showing the effect of GATAD2B knockdown in a wild- type KRAS background cell line as for the data shown in TCGA human data analysis as well as in vivo tumor experiment as in 3D (GATAD2B off condition) brings the indication in that direction.

Response: Upon reflection of the reviewer's comment describing a GATAD2B off condition in Fig. 3D, we realized the unclear manner in which we described the conditions tested in this model. We have updated our results to reflect this clarification, stating the inducible HBEC *iKRAS* cells activate *KRAS*^{G12D} only, and only in the presence of doxycycline. **The *GATAD2B* or *GFP* control expression mentioned in this manuscript is either constitutively present or absent as described.** Therefore, when On Dox or Off Dox is designated, only the expression of *KRAS*^{G12D} is being manipulated.

To demonstrate the cooperativity between *KRAS*^{G12D} and *GATAD2B*, we describe several approaches in this manuscript outlined below:

1. Transwell invasion including four conditions provided in **Fig. 3C**, where *in vitro* results indicate *KRAS*^{G12D} activation alone induces a slight elevation in invasion as expected (left-hand side, comparing GFP;off Dox vs. GFP;on Dox). When *GATAD2B* is constitutively expressed (right side), the invasive phenotype is robustly observed only under the induction of doxycycline (*KRAS*^{G12D}) and not in the absence of *KRAS*^{G12D}, indicated in the Off Dox condition (3rd bar in the bar graph).
2. In a similar manner, we have demonstrated the *in vivo* cooperativity between *GATAD2B* and *KRAS*^{G12D}. In the presence of doxycycline (*KRAS*^{G12D} activation), tumor growth is robustly demonstrated only in the presence of exogenous

GATAD2B (red and purple lines of **Fig. 3D**), which is promptly abrogated upon removal of mutant *KRAS*, suggesting *KRAS*^{G12D} and *GATAD2B* are cooperating to drive the tumor growth phenotype.

- To determine whether *GATAD2B* is required for anchorage independent colony growth in *KRAS* mutant cancer, we analyzed *GATAD2B* depletion in *KRAS* mutant NSCLC cell lines. Upon *GATAD2B* knockdown, we observe a significant decrease in the cells ability to form colonies. **Thanks to the reviewer's excellent suggestion to test a *KRAS* wild-type background, we have included these data in which we tested *KRAS* wild-type NSCLC cell lines (H1437 and H1568).** There were no observable differences in anchorage independence upon *GATAD2B* knockdown in the *KRAS* wild-type NSCLC tested (**Fig. 3G**). We thank the reviewer for this suggestion and have added this important control to **Fig. 3G**, as seen below.

5. Figure 3D – to my opinion the authors have focused on *GATAD2B* results to prove their point, which is fine. But on the chart, the lines that follow ‘GFP on’ vs. ‘GFP on>off’ has been ignored. The GFP result has also shown similar trend for on>off experiment, which complicate the claim that the authors are drawing here. Of course for the GFP lines the tumors are half the size than the *GATAD2B* lines and therefore the drop is hidden, but when one compare ‘On’ to ‘On>off’ situation for both GFP and *GATAD2B*, the trend in tumor size reduction over time is similar or same. Please explain with additional proof.

Response: We thank the reviewer for their careful review of **Fig. 3D**. As this panel is also dependent on the inducible cell model, we hope the revised manuscript with improved clarity of the model, will address the conclusion the reviewer has reached on this comment as well. The *KRAS*^{G12D};GFP and *KRAS*^{G12D};GFP ON→OFF controls the reviewer mentions illustrate that *KRAS*^{G12D} is necessary and sufficient to drive initial HBEC tumorigenesis. Interestingly, we only observe small palpable tumors that remain stably and reliably small when doxycycline is continually administered (**Fig 3D, blue**). The reviewer is correct that doxycycline withdrawal will not result in a continued presence of these small tumors (**Fig. 3D, green**). In fact, we expect the small reduction

observed here, given that $iKRAS^{G12D}$ HBEC cells, without doxycycline ($KRAS^{wt}$ condition) do not develop tumors (**Fig 3D, orange**). This observation is consistent with previously published reports in HBECs, where loss of oncogenic $KRAS$ results in loss of tumor growth (PMID: 21306997). Our goal was to test the hypothesis that $GATAD2B$ expression requires $KRAS$ for oncogenic transformation and progression. Moreover, given the striking tumor growth phenotype observed in these mice, we sought to determine whether mutant $KRAS$ was required for the rapid growth of the $GATAD2B$ cohort since $KRAS^{G12D}$ alone does not result in the rapid tumor growth phenotype (**Fig. 3D, green**). Importantly, we also do not observe a tumor growth phenotype in $GATAD2B$ alone conditions, as evidenced by the lack of tumor formation in the $GATAD2B$;Off Dox cohort (absence of $KRAS^{G12D}$ activation, **Fig. 3D black**).

Here we have demonstrated that the growth and maintenance of these tumors is dependent on both $KRAS^{G12D}$ and $GATAD2B$, as withdrawal of $KRAS^{G12D}$ results in significant tumor regression (**Fig. 3D, purple**) and this growth is only observed upon addition of $GATAD2B$ in $KRAS^{G12D}$ HBECs. Taken together, the data suggests they are both required together to drive tumor growth.

To provide additional proof of our assertion for this cooperation, we determined whether the proliferation rate in the $GATAD2B$ ON→OFF tumor cohort was higher than the GFP ON→OFF cohort. Both cohorts were analyzed after re-introduction of doxycycline-induced $KRAS^{G12D}$ activation. Consistent with our tumor growth assay, we observed that $GATAD2B + KRAS^{G12D}$ activation led to significantly higher levels of the proliferation marker Ki-67 compared with addition of $KRAS^{G12D}$ only, consistent with the findings in Fig. 3E.

6. The proliferation data shows a reduced proliferation of GFP and GATAD2B cells, on induction with dox as compared to dox negative condition, which is contradictory to the hypothesis (Fig supp 4A). Perhaps, this data could be validated by a much more direct approach of measuring proliferation like EdU/BrDU assay.

Response:

We thank the reviewer for highlighting an important observation we had not previously explained. We agree that describing the proliferation phenotype in HBECs improves our argument that the addition of *GATAD2B* plays a significant role in invasion without altering proliferation levels *in vitro* and during the time course of our invasion assay. In fact, the HBEC proliferation phenotype is only altered by *KRAS*^{G12D} activation.

To first address the reviewer's concern regarding a direct approach to measuring proliferation, in which we describe the use of the CellTiter-Glo assay, we clarify here that we and several others have previously published studies using this assay to directly measure proliferation changes (PMID: 23302800, PMID: 26806015, PMID: 23051747, PMID: 20730488, PMID:7680699) through quantitation of the proportional amount of ATP produced by viable cells, and we have updated our methods and references section to improve the justification of this assay.

The reviewer has also made an excellent observation that there is a significantly lower rate of proliferation upon addition of *KRAS*^{G12D}, and we fully agree this is an additional phenotypic change that was not explained in our initial manuscript. Based on previous reporting in HBECs (PMID:23449933, PMID:9054499), we hypothesized that the reduced proliferation phenotype is the result of using a non-transformed cell line that may undergo a degree of oncogene-induced senescence. Given Reviewer #2 also reached this hypothesis (comment 4), we stained our HBECs for the senescence marker, beta-galactosidase. We observed a significant increase in the number of senescent cells in the presence of *KRAS*^{G12D}, to a similar degree as was observed using a senescence-inducing agent, etoposide (see figure below, measured according to manufacturer's instructions). We thank the reviewer for their thoughtful observation, and we have updated the results and methods sections, as well as **Supp. Fig. 4** to include the results included below to explain this observation:

7. In supplementary figure 4C, it mentioned that there is no change in pERK in presence or absence of dox, but from the immunoblot it is quite clear that there is an increase in pERK2(upper band), with dox treatment, in both GFP and GATAD2B cells. Whereas the pERK1, is only increased with dox in GFP control cells and remains unaltered in GATAD2B cells. What could be the probable justification for this result?

Response: Reflecting on the reviewer's comment, we determined the reviewer's conclusion may also be a result of the unclear manner in which we described our inducible cell model, and therefore clarification of the model in the updated results section may help to address the conclusion this reviewer has reached. The statement in the text (Line# 244) indicates that *GATAD2B* did not influence MAPK signaling (Lane 3, *GATAD2B* alone without dox-induced *KRAS*^{G12D}). In fact, the presence of doxycycline-induced *KRAS*^{G12D} expression should be expected to influence pERK signaling, and all samples with addition of doxycycline resulted in elevated levels of pERK2 as expected. In this experiment, we sought to demonstrate that addition of exogenous, constitutive, *GATAD2B* does not enhance MAPK signaling on its own or in combination with the inducible mutant *KRAS*. A result in which *GATAD2B* were influencing MAPK signaling would be demonstrated by a darker band in the *GATAD2B* + Dox lane compared to the GFP + Dox, which we do not observe. We feel this comment again reflects our miscommunication regarding the effects of the induction of doxycycline, which only activates *KRAS*^{G12D} expression, and we hope the clarifications we have included in the results section properly conveys the experimental conditions.

8. Fig 4E - What is the 110 KD loading control used and why? The sample loading show so much variability? Tubulin or actin loading control should have been ideal for this

case. Considering everything gone right as presented here, authors need to explain why is it that p53 expression is more in proximal lymph nodes, than the lungs?

Response: We appreciate the reviewer's thoughtful concerns with sample loading variability observed in **Fig. 4E**. These immunoblots contain lysates extracted from *in vivo* tumors, in which a standard loading control was used (Vinculin, CST #4650). We thank the reviewer for noting the absence of this information, and we have updated our methods section to include use of this control. However, as with many extracted murine tissue types, we can expect variability in protein levels between different tissue types. Despite this variation, we reasoned that differences in vinculin between cell types does not affect the conclusion reached in our findings. Rather than to compare p53 expression levels between lung and lymph node, our goal with this data is to demonstrate our technique as a *bona fide* model, and we therefore sought to determine whether exogenous human p53 was observed in the lungs of mice intubated with high titer p53-containing lentivirus. In fact, as demonstrated in the western blot and below from quantification of the protein levels, we observe marked differences in p53 levels in the lungs of mice treated with p53-cre lentivirus compared with mice treated with a GFP-cre lentivirus, a finding which was consistent with previously published reports cited in our manuscript (PMID: 19561589).

The reviewer also raises concerns regarding levels of p53 in lymph nodes as higher than levels present in the lung. Reflecting on this comment, we realized a potentially confusing and misaligned angle in **Fig. 4E**, which has been updated in our figure set and included below. Levels of p53 in *GFP-Cre* lymph node were not presented due to lack of appreciable material collection from normal lymph node size in murine models. As observed in **Fig. 4C**, lymph node with tumor infiltration may contain a large amount of tumor cells, which we would expect to be driven by p53, therefore we immunoblotted for p53 levels.

9. Figure 4G & H – In the context of Cre and GATAD2B lentivirus experiment, as claimed in discussion (line 388-389), the role of GATAD2B in driving tumor progression and ‘METASTASIS’ is overstatement. These experiments only show that GATAD2B lentivirus can enhance tumorigenesis. Further, additional results and discussion is required explaining whether and how this GATAD2B-cre lentivirus create lung specific higher luciferase signal.

Response:

Subcomment1: We thank the Reviewer for their comment and agree the statement is overdrawn in the Cre lentivirus experiment. As we have not observed metastases at the time points presented in this autochthonous model, we have revised the Results and Discussion to clarify that *GATAD2B* expression in the *LSL-KRAS^{G12D}* model drive lung tumor growth, but not metastasis.

Subcomment 2: The lung specific luciferase signal is a result of tumor growth in the conditionally-activated luciferase-Cre mice. We appreciate the reviewer providing us an opportunity to clarify these results. Upon addition of lentivirus, delivered directly into the lungs of mice, the luciferase activity will be activated in both *GFP-Cre* and *GATAD2B-Cre* cohorts when tumors develop as a result of oncogenic activation. In these animals the luciferase signal approximately correlates with tumor burden, as can be appreciated by the luciferase images with time-matched *GFP;KRAS^{G12D}* and *GATAD2B; KRAS^{G12D}*-driven lung tumors included below, which we have now added as **Supp. Fig. 6**:

As you can appreciate from these data, the luciferase signal increases over time, which we hypothesized would indicate increased tumor burden. Upon necropsy, we observe much larger tumors in the *GATAD2B*-treated mice, consistent with observed luciferase signal. We thank the reviewer for suggesting additional discussion to clarify this point and we have updated the results section to reflect this.

10. Figure 5A – What has been represented in the X-axis is not clear. BRET methodology on page 19 also do not explain how the BRET ratio is calculated – is it a simple Acceptor/donor ratio represented in Y-axis or is this bleedthrough corrected BRET ratio represented as ‘Net BRET’ on that axis. In any case, X-axis representation needs to be clearly mentioned. Further, what does it mean ‘FOCAUC>1.0’ on line 323. Authors should explain all abbreviation at the first occurrence as standard.

Response: We appreciate the reviewer’s important points that are critical for the clarity of Figure 5A. To address reviewer’s concerns regarding the meaning of X- and Y-axis, and abbreviation, we have revised the results and methods section accordingly. Specifically, it is now indicated on page 12 that “*Statistical analysis of BRET signal, protein expression signal and BRET_n saturation curves were performed to define positive PPIs with fold-over control of the area under curve (FOCAUC) >1.0, p<0.01.*” Now it is also explained in the “*GATAD2B interactors screening*” (page 20) that, “*The BRET ratio is calculated as the ratio of intensity at 535 nm over 460 nm. The net BRET is the bleed-through corrected BRET ratio. The relative amount of NLuc-GATAD2B and Venus-gene partners expressions were measured by the luminescence signal at 460 nm (L_{460}) and the laser-excited Venus fluorescence intensity (FI). The ratio of relative amount of acceptor over donor protein expression (Acceptor/Donor) was defined as Venus FI/ L_{460} . Area under the BRET saturation curve (AUC) was built by fitting the net BRET on Y-axis and Acceptor/Donor on X-axis using one-site binding equation, and the*

statistical significance was calculated with the unpaired two-tailed t-test as described previously (Mo and Fu, 2016; Mo et al., 2016)."

Reviewer #2 (Reviewer Comments to the Author):

Grzeskowiak and colleagues describe an *in vivo* gain of function screen for metastasis drivers in NSCLC. Starting from transcriptome profiles derived from comparing primary NSCL cell lines and corresponding metastases in combination with mouse/human cross-species comparisons, they defined a set of 251 genes potentially associated with metastasis. They used this gene list for an *in vivo* gain of function screen. The study then validates and functionally characterizes Gatad2d, one of the candidate discovered in the screen. The principal findings of the study are (1) identification of novel metastasis-promoting oncogenes (28 genes emerged from this screen), (2) cross-species mouse/human comparisons to confirm the potential human relevance, (3) validation of Gatad2b as a bona-fide context-dependent (Kras mutant NSCLC) metastasis-promoting oncogene, and (4) identification of Myc as a Gatad2b interaction partner and a downstream mediator of Gatad2b-dependent effects on tumor growth and metastasis.

The screen using barcoded expression vectors is innovative and the results are persuasive. The manuscript is well written and the results are discussed in a balanced manner. Overall this is an excellent study, which in my opinion is suitable for publication in Nature communications. It not only provides a list of metastasis drivers (which can in future be picked up by others) but also a convincing validation of Gatad2b and mechanistic insights into possible downstream effects.

Response: We thank the reviewer for recognition of the innovative and persuasiveness of our study. We address the Reviewer's comments below.

1. It seems that human ORFs are used in murine cancer cells (page 5, 2nd paragraph). Some ORFs might therefore not be functional in the screen (giving false negative results). Can the authors comment on this issue?

Response: We appreciate the reviewer's recognition of potential false negative results arising from our *in vivo* primary screening model. As the reviewer correctly points out, we have used human ORFs in this study in murine cancer cells. We chose to integrate human ORF collections into this system given there are no mouse ORF collections commercially available for this scale of functional studies. We are encouraged by the fact that morphology between cancers arising from this model are shared and oncogenic genes are often highly conserved between human and mouse (PMID:

10688857). This approach has been successfully integrated into several previously published reports to yield novel oncogenic drivers by others (PMID: 27478040, PMID: 27899381, PMID: 27320920). However, we acknowledge there is the potential for false negatives in the case where human and mouse genomes may not share conservation. To test conservation in our candidate list, we used the NCBI Homologene database, which compares BLAST sequences in a pair-wise manner between mouse and human. This analysis demonstrated all of the candidates tested in our screening strategy were highly conserved between mouse and human. We thank the reviewer for this insight, and we have updated our results section and included this analysis in **Supp.Table 1**.

2. Please comment on the use of wild type ORFs for oncogenes altered by activating mutations in human tumours (page 5, 2nd paragraph).

Response: We thank the reviewer for their careful consideration of the methodology used to generate a gene candidate list for functional screening. We understand the reviewer would like additional rationale for substituting wild-type ORFs for activating mutations. While we recognize this is not true for all oncogenic drivers, we reasoned that oncogenic pathways amplified through copy number-driven overexpression can be similarly activated to hyperactivation mutations, as has been previously reported in notable oncogenes including *PIK3CA* and *ALK* (PMID: 9916799, PMID: 15016963, PMID: 15972965, PMID: 18923525, PMID: 18724359). However, we acknowledge there is a possibility that exogenous expression via wild-type ORF does not recapitulate an oncogenic event. However, given a number of scoring ORFs were obtained from Ding et al. mutation data (*GNAS*, *MAPK6*, *ACVR1B*, *FBXW7*, *NTRK3*), the overexpression of these genes merit further interrogation for future studies. We have updated our results section to reflect these comments.

3. Figure 1A: Annotation of promoter (EF1a?) in vector map is missing.

Reponses: We thank the reviewer for their attentiveness to this detail and agree this will improve our figure. We have updated **Fig. 1A** to include annotation of the promoter used in our primary screening model.

A

4. Supplementary Figure 4A: *Kras*-expressing cells (GFP + DOX and GATA + DOX) seem to proliferate markedly slower than *Kras* wild type cells (GFP – DOX and GATA – DOX), which is counterintuitive. Is there evidence for senescence? Please comment.

Response: We completely agree with the reviewer regarding the counterintuitive nature of our proliferation phenotype. As the reviewer has thoughtfully commented, we too hypothesized an oncogene-induced senescence. We appreciate the importance of this observation, as it was also brought up by reviewer #1, Comment #6. In this figure we are using a non-transformed cell model for which others have reported a similar phenotypic observation (PMID:23449933, PMID:9054499). We stained our HBECs for the senescence marker, beta-galactosidase, and observed a significant increase in the number of senescent cells in the presence of *KRAS*^{G12D}, to a similar degree as was observed using a senescence-inducing agent, etoposide. These new data are provided in **Supp. Fig. 4B** of our revised manuscript.

Reviewer #3 (Reviewer Comments to the Author):

This manuscript reports on a the use of in vitro and in vivo functional genetics platforms to identify mediators of lung cancer metastasis. The system and the findings are interesting and the experiments are well conducted. There are issues to address before firm conclusion can be drawn.

Response: We thank the Reviewer for their comments on our experimental design. We address the reviewer's comments below.

(1) The authors use a flank system to identify mediators of lung cancer metastasis. This system is plagued by the fact that the findings are not derived from a spontaneous lung cancer model. This needs to be experimentally addressed both for GATA2B and MYC.

Response: We fully agree with the reviewer's point that the most robust model to characterize a mediator of lung cancer metastasis is with a spontaneous lung cancer model. The initial goal of this study was to functionally screen a large number of candidate genetic aberrations obtained from oncogenomics-informed datasets to identify the most promising drivers of metastasis through *in vivo* means by providing biological context. For a streamlined screening strategy, we chose to screen for metastatic lesions derived from subcutaneous primary tumors, as this facilitates the monitoring of primary tumor growth and recovery of control/input tissues for sequencing. This screening strategy produced a list of 28 potential genes for further validation/characterization. We acknowledge that the functional assessment of any gene scoring as a metastasis driver in this primary screening strategy needs validation

with additional evidence to demonstrate malignant phenotypes, and we have updated our Discussion section to reflect this critical point.

We agree that transgenic *Gatad2b* animal models eliciting spontaneous lung cancer metastasis is an excellent way to demonstrate these capabilities, and development of models of this magnitude are the subject of future grant proposals. However, the progression of spontaneous metastasis in mouse models under development would require additional experiments well beyond the time allotted by the Editors. We hope the reviewer is convinced the feasibility of this experiment exceeds the scope of this paper, in which *GATAD2B* has been identified as a pro-tumorigenic and pro-metastatic driver from an *in vivo* genetic screen, validated through *in vitro* phenotypic characterization demonstrating invasiveness, and *in vivo* using both human bronchial epithelial cells and the lenti-Cre-mediated generation of *KRAS*^{G12D};*GATAD2B* expressing lung tumors.

Regarding the reviewers comment on producing a spontaneous metastasis model driven by MYC, we agree with the reviewer and we have updated our Discussion section to cite published studies of murine models demonstrating spontaneous metastasis and enhanced tumor progression in *KRAS*^{G12D}-driven tumors (PMID: 19551151, PMID:18356293), in addition to human studies correlating elevated MYC levels with worse clinical outcomes, disease progression, and metastasis (PMID: 8100491, PMID: 23431158, PMID: 22469662). Generating a spontaneous lung cancer metastasis model of MYC in our lab would not produce a novel finding, as these models have been previously published. We hope the reviewer agrees that others have adequately demonstrated MYC as a driver of metastasis in NSCLC.

(2) The mechanisms of cooperativity between KRAS and MYC is not shown. This is a major gap.

Response: We acknowledge the reviewer's concern for cooperativity between MYC and KRAS. Studies demonstrating cooperativity between MYC and KRAS are not novel and outside of the scope of this work. Poor prognostic outcomes for the combination of mutant *KRAS* and amplified *MYC* have been well established in human cancers (PMID: 23555992, PMID: 25079552), with mouse models to demonstrate metastatic mechanisms of *KRAS*^{G12D} and elevated *MYC* (PMID: 23431158). Moreover, this work has also already been demonstrated in HBEC models, where *MYC* greatly enhanced aggressiveness of *KRAS* neoplastic transformation and induction of epithelial-to-mesenchymal transition (EMT) in HBECs (PMID: 23449933 PMID: 27500490). We hope the reviewer agrees a mechanistic study between KRAS and MYC is outside the scope of this project, where the primary goal and novelty of our work, as highlighted by reviewer #1, comment 1, was identification of genes capable of driving metastasis *in vivo*, where *GATAD2B* was identified and validated as a driver of tumor progression and metastasis. We have refocused our Discussion section to better address future work

aimed at understanding KRAS and MYC cooperativity in the context of GATAD2B, and included additional publications of work previously completed demonstrating KRAS and MYC cooperativity.

(3) The authors use KRAS mutant models but fail to show whether the axis identified is specific for KRAS biology or non-specific.

Response: We thank the reviewer for highlighting this important point and we fully agree demonstrating *GATAD2B*-driven phenotypes in *KRAS* mutant and *KRAS* wild-type conditions strengthens our hypothesis that *GATAD2B* and *KRAS*^{G12D} work together in lung cancers to drive pro-tumorigenic and pro-metastatic activity. To determine whether *GATAD2B* is required for anchorage independent colony growth in *KRAS* mutant cancer, we analyzed *GATAD2B* depletion in *KRAS* mutant NSCLC cell lines. Upon *GATAD2B* knockdown, we observe a significant decrease in the cells ability to form colonies. **Thanks to the reviewer's excellent suggestion to test a *KRAS* wild-type background**, we have included these data in which we tested *KRAS* wild-type NSCLC cell lines (NCI-H1437 and NCI-H1568). There were no observable differences in anchorage independence upon *GATAD2B* knockdown in the *KRAS* wild-type NSCLC tested (**Fig. 3G**). We thank the reviewer for this suggestion and have added this important control to Fig. 3G, as seen below.

(4) The role of the signaling axis identified in tumor growth versus metastatic processes remains unclear and should be mechanistically elucidated before drawing firm conclusions.

Response:

In consideration of the Reviewer's comment, we fully agree that more work needs to be done to demonstrate cooperativity mechanisms between oncogenic and metastatic drivers. This study herein describes a functional screening platform to readily identify novel candidates that have now been validated and poised for future studies. Our

results indicate a subset of our metastasis drivers also scored in the primary tumor (**Supp. Fig. 2B**), suggesting that genes that drive tumor progression and aggression are also found in metastatic lesions. In our follow up work with *GATAD2B*, we identified that these actions activate MYC target pathways, which as previously described, drive tumor aggression. Other pathways identified by RPPA, (**Supp. Table 5, Supp. Fig. 6, Fig. 5E**) are also commonly shared between tumor growth drivers and metastasis-inducing genes. These include HIF1A and hypoxia (PMID: 29339479), MET pathway activation (PMID: 28989054), and mTOR signaling (PMID: 22367541). We hope the reviewer agrees the complexity of the studies required to follow-up these various independent pathway alterations warrants an entirely separate manuscript and is the subject of new grant proposals based on this work. We have restructured our discussion to better address the novelty and findings in this paper, including the *in vivo* metastasis screen and the identification of the role of *GATAD2B* in enhancing tumor growth and metastasis *in vitro* and *in vivo*, clarifying the scope of our work, and the work of future studies.

REVIEWERS' COMMENTS:

Reviewer #1 (Remarks to the Author):

Revised manuscript has satisfactorily addressed all major concerns. No further suggestion.

Reviewer #2 (Remarks to the Author):

My questions and suggestions have been addressed. I recommend publication.

Reviewer #3 (Remarks to the Author):

The authors have submitted a revised manuscript. While the in vivo screening platform is interesting, the strength and depth of the mechanistic insight in this revised manuscript still falls short of what is necessary for publication in Nature Communications. Specifically, the authors provide little if any new experimental data to address points #1, #2, and #4 in my original review. I believe the authors findings would likely have been significant enough to warrant publication Nature Communications, if these points were addressed experimentally. Without such information, the manuscript is more of a technical report suitable for a more cancer-focused or technical journal.

Response to reviewers
Nature Communications
NCOMMS-17-19578A

In Vivo Screening Identifies GATAD2B as a Metastasis Driver in Kras-Driven Lung Cancer

REVIEWERS' COMMENTS:

Reviewer #1 (Remarks to the Author):

Revised manuscript has satisfactorily addressed all major concerns. No further suggestion.

Response: We thank the reviewer for their comments and appreciate their effort to improve our manuscript.

Reviewer #2 (Remarks to the Author):

My questions and suggestions have been addressed. I recommend publication.

Response: We thank the reviewer for their comments and appreciate their effort to improve our manuscript.

Reviewer #3 (Remarks to the Author):

The authors have submitted a revised manuscript. While the in vivo screening platform is interesting, the strength and depth of the mechanistic insight in this revised manuscript still falls short of what is necessary for publication in Nature Communications. Specifically, the authors provide little if any new experimental data to address points #1, #2, and #4 in my original review. I believe the authors findings would likely have been significant enough to warrant publication Nature Communications, if these points were addressed experimentally. Without such information, the manuscript is more of a technical report suitable for a more cancer-focused or technical journal.

Response: We thank the reviewer for noting interest in our *in vivo* screening platform. Beginning with 250 oncogenomically-informed candidates potentially driving metastasis in *KRAS*-driven lung cancer, our work presented here has narrowed a focus on aggressive candidates in mouse and human models. From the 28 identified in our screen, there are many interesting candidates that may be of interest to others in the field. In this manuscript and the co-submitted manuscript (Kundu et al.), we have outlined the mechanistic roles for two of the drivers identified in the screen. Herein, using patient data available from TCGA and experimentally, we identified a synergistic relationship between *GATAD2B* and mutant *KRAS*. We demonstrate pro-tumorigenic and pro-metastatic activity *in vivo* and *in vitro*, employing the use of non-transformed human bronchial epithelial cells (HBECs) and using autochthonous mouse models of lung cancer. Moreover, we mechanistically show a physical interaction between *GATAD2B* and *MYC*, which is a gene shown in multiple previously published studies of *KRAS*-driven cancer to drive aggressive phenotypes. We share the reviewers appetite to better understand mechanistically how these critical oncogenic pathways synergize to promote progression and metastasis, but there are certainly limits to how far this work can be extended in any one manuscript. While we feel some of the recommended studies are outside the scope of the current manuscript, we hope to expand on these studies for deeper mechanistic insights in future work.